# ComPhy: Compositional Physical Reasoning of Objects and Events from Videos

**Zhenfang Chen**
MIT-IBM Watson AI Lab

**Kexin Yi**
Harvard University

**Yunzhu Li**
MIT

**Mingyu Ding**
The University of Hong Kong

**Antonio Torralba**
MIT

**Joshua B. Tenenbaum**
MIT BCS, CBMM, CSAIL

**Chuang Gan**
MIT-IBM Watson AI Lab

## Abstract

Objects' motions in nature are governed by complex interactions and their properties. While some properties, such as shape and material, can be identified via the object's visual appearances, others like mass and electric charge are not directly visible. The compositionality between the visible and hidden properties poses unique challenges for AI models to reason from the physical world, whereas humans can effortlessly infer them with limited observations. Existing studies on video reasoning mainly focus on visually observable elements such as object appearance, movement, and contact interaction. In this paper, we take an initial step to highlight the importance of inferring the hidden physical properties not directly observable from visual appearances, by introducing the Compositional Physical Reasoning (ComPhy) dataset [1]. For a given set of objects, ComPhy includes *few* videos of them moving and interacting under different initial conditions. The model is evaluated based on its capability to unravel the compositional hidden properties, such as mass and charge, and use this knowledge to answer a set of questions posted on one of the videos. Evaluation results of several state-of-the-art video reasoning models on ComPhy show unsatisfactory performance as they fail to capture these hidden properties. We further propose an oracle neural-symbolic framework named Compositional Physics Learner (CPL), combining visual perception, physical property learning, dynamic prediction, and symbolic execution into a unified framework. CPL can effectively identify objects' physical properties from their interactions and predict their dynamics to answer questions.

## 1 Introduction

Why do apples float in water while bananas sink? Why do magnets attract each other on a certain side and repel on the other? Objects in nature often exhibit complex properties that define how they interact with the physical world. To humans, the unraveling of new intrinsic physical properties often marks important milestones towards a deeper and more accurate understanding of nature. Most of these properties are *intrinsic* as they are not directly reflected in the object's visual appearances or otherwise detectable without imposing an interaction. Moreover, these properties affect object motion in a *compositional* fashion, and the causal dependency and mathematical law between different properties are often complex.

As shown in Fig. 1, different intrinsic physical properties, such as charge and inertia, often lead to drastically different future evolutions. Objects carrying the same or opposite *charge* will exert a repulsive or attractive force on each other. The resulting motion not only depends on the amount of charge each object carries, but also their signs (see Fig. 1-(a)). The *inertia* determines how sensitive an object's motion is to external forces. When a massive object interacts with a light object via attraction, repulsion, or collision, the lighter object will undergo larger changes in its motion compared with the massive object's trajectory (see Fig. 1-(b)).

Recent studies have established a series of benchmarks to evaluate and diagnose machine learning systems in various physics-related environments (Bakhtin et al., 2019; Yi et al., 2020; Baradel et al.,

---

[1]Project page: https://comphyreasoning.github.io

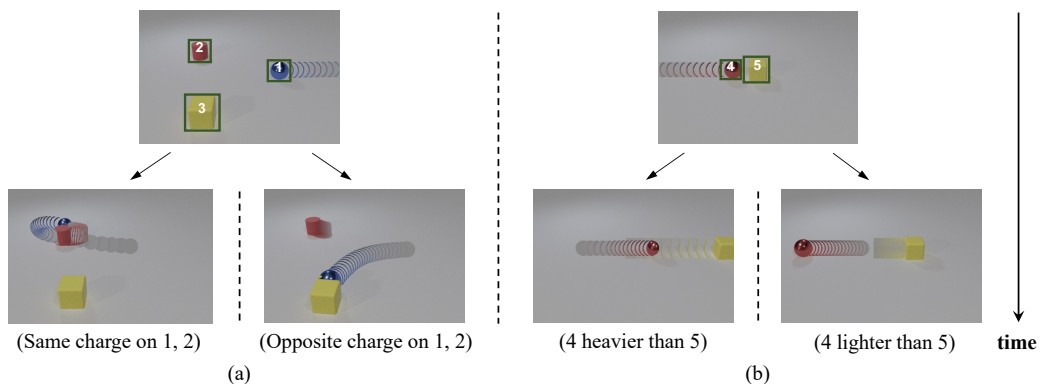

Figure 1: Non-visual properties like mass and charge govern the interaction between objects and lead to different motion trajectories. a) Objects attract and repel each other according to the (sign of) charge they carry. b) Mass determines how much an object's trajectory is perturbed during an interaction. Heavier objects have more stable motion.

2020). These benchmarks introduce reasoning tasks over a wide range of complex object motion and interactions, which poses enormous challenges to existing models since most tasks require them to fully capture the underlying physical dynamics and, in some cases, be able to make predictions. However, the majority of complexity in the motion trajectories facilitated by these environments comes from changes or interventions in the initial conditions of the physical experiments. The effects of object intrinsic physical properties, as well as the unique set of their challenges, are therefore of great importance for further investigation.

It's non-trivial to build a benchmark for compositional physical reasoning. Existing benchmarks (Yi et al., 2020; Ates et al., 2020) assume that there is no variance in objects' physical properties and ask models to learn physical reasoning from massive videos and question-answer pairs. A straightforward solution is to correlate object appearance with physical properties like making all *red spheres* to be *heavy* and then ask questions about their dynamics. However, such a design may incur shortcuts for models by just memorizing the appearance prior rather than understanding coupled physical properties. In this work, we propose a novel benchmark called ComPhy that focuses on understanding object-centric and relational physics properties hidden from visual appearances. ComPhy first provides few video examples with dynamic interactions among objects for models to identify objects' physical properties and then asks questions about the physical properties and corresponding dynamics. As shown in Fig. 2, ComPhy consists of meta-train sets and meta-test sets, where each data point contains 4 reference videos and 1 target video. Within each set, the objects share the same intrinsic physical properties across all videos. Reasoning on ComPhy requires the model to infer the intrinsic and compositional physical properties of the object set from the reference videos, and then answer questions about this query video. To make the task feasible, we systematically control each object in the query video that should appear at least in one of the reference videos.

We also introduce an oracle model to tackle this task. Inspired by recent work on neural-symbolic reasoning on images and videos (Yi et al., 2018; 2020; Chen et al., 2021), our model consists of four disentangled components: perception, physical property learning, dynamics prediction, and symbolic reasoning. Our model is able to infer objects' compositional and intrinsic physical properties, predict their future, make counterfactual imaginations, and answer questions.

To summarize, this paper makes the following contributions. First, we present a new physical reasoning benchmark ComPhy with physical properties (mass and charge), physical events (attraction, repulsion), and their composition with visual appearance and motions. Second, we decorrelate physical properties and visual appearance with a few-shot reasoning setting. It requires models to infer hidden physical properties from only a few examples and then make predictions about the system's evolution to help answer the questions. Third, we propose an oracle neural-symbolic framework, which is a modularized model that can infer objects' physical properties and predict the objects' movements. At the core of our model are graph neural networks that capture the compositional nature of the underlying system.

## 2 RELATED WORK

**Physical Reasoning.** Our work is closely related to recent developments in physical reasoning benchmarks (Riochet et al., 2018; Girdhar & Ramanan, 2020; Ates et al., 2020; Hong et al., 2021). PHYRE (Bakhtin et al., 2019) and its variant ESPRIT (Rajani et al., 2020) defines an environment

| Dataset | Video | Question Answering | Diagnostic Annotation | Composition | Few-shot Reasoning | Physical Property | Counterfactual Property Dynamics |
|---|---|---|---|---|---|---|---|
| CLEVR (Johnson et al., 2017) | - | ✓ | ✓ | ✓ | - | - | - |
| MovieQA (Tapaswi et al., 2016) | ✓ | ✓ | - | ✓ | - | - | - |
| TGIF-QA (Jang et al., 2017) | ✓ | ✓ | - | - | - | - | - |
| TVQA/ TVQA+ (Lei et al., 2019) | ✓ | ✓ | - | ✓ | - | - | - |
| AGQA (Grunde-McLaughlin et al., 2021) | ✓ | ✓ | - | - | - | - | - |
| IntPhys (Riochet et al., 2018) | ✓ | - | ✓ | - | - | ✓ | - |
| PHYRE/ ESPRIT (Rajani et al., 2020) | ✓ | - | ✓ | ✓ | - | ✓ | - |
| Cater (Riochet et al., 2018) | ✓ | ✓ | ✓ | ✓ | - | - | - |
| CoPhy (Baradel et al., 2020) | ✓ | - | ✓ | - | - | ✓ | - |
| CRAFT (Ates et al., 2020) | ✓ | ✓ | ✓ | ✓ | - | - | - |
| CLEVRER (Yi et al., 2020) | ✓ | ✓ | ✓ | ✓ | - | - | - |
| **ComPhy (ours)** | ✓ | ✓ | ✓ | ✓ | ✓ | ✓ | ✓ |

Table 1: Comparison between ComPhy and other visual reasoning benchmarks. ComPhy is the dataset with reasoning tasks for both physical property learning and corresponding dynamic prediction.

where objects can move within a vertical 2D plane under gravity. Each task is associated with a goal state, and the model solves the task by specifying the initial condition that will lead to the goal state. CLEVRER (Yi et al., 2020) contains videos of multiple objects moving and colliding on a flat plane, posting natural language questions about description, explanation, prediction, and counterfactual reasoning on the collision events. CoPhy (Baradel et al., 2020) includes experiment trials of objects moving in 3D space under gravity. The task focuses on predicting object trajectories under counterfactual interventions on the initial conditions. Our dataset contributes to these physical reasoning benchmarks by focusing on physical events driven by object intrinsic properties (situations shown in Fig. 2). ComPhy requires models to identify intrinsic properties from only a few video examples and make dynamic predictions based on the identified properties and their compositionality.

**Dynamics Modeling.** Dynamics modeling of physical systems has been a long-standing research direction. Some researchers have studied this problem via physical simulations, drawing inference on the important system- and object-level properties via statistical approaches such as MCMC (Battaglia et al., 2013; Hamrick et al., 2016; Wu et al., 2015), while others propose to directly learn the forward dynamics via neural networks (Lerer et al., 2016). Graph neural networks (Kipf & Welling, 2017), due to their object- and relation-centric inductive biases and efficiency, have been widely applied in forwarding dynamics prediction on a wide variety of systems (Battaglia et al., 2016; Chang et al., 2016; Sanchez-Gonzalez et al., 2020; Li et al., 2019a). Our work combines the best of the two approaches by first inferring the object-centric intrinsic physical properties and then predicting their dynamics based on the intrinsic properties. Recently, VRDP (Ding et al., 2021) performs object-centric dynamic reasoning by learning differentiable physics models.

**Video Question Answering.** Our work is also related to answering questions about visual content. Various benchmarks have been proposed to handle the tasks of cross-modal learning (Lei et al., 2018; Chen et al., 2019; Li et al., 2020; Wu et al., 2021; Hong et al., 2022). However, they mainly focus on understanding human actions and activities rather than learning physical events and properties, which is essential for robot planning and control. Following CLEVRER, we summarize the difference between ComPhy and previous benchmarks in Table 1. ComPhy is the only dataset that requires the model to learn physical property from few video examples, make dynamic predictions based on the physical property, and finally answer corresponding questions.

**Few-shot Learning.** Our work is also related to few-shot learning, which typically learns to classify images from only a few labelled examples (Vinyals et al., 2016; Snell et al., 2017; Han et al., 2019). ComPhy also requires models to identify objects' property labels from only a few video examples. Different from them, reference videos have no labels for objects' physical properties but more interaction among objects, providing information for models to identify objects' physical properties.

## 3 DATASET

ComPhy studies objects' *intrinsic* physical properties from objects' interactions and how these properties affect their motions in future and counterfactual scenes to answer corresponding questions. We first introduce videos and task setup in Section 3.1. We then discuss question types in Section 3.2, and statistics and balancing in Section 3.3.

### 3.1 VIDEOS

**Objects and Events.** Following Johnson et al. (2017), objects in ComPhy contain compositional appearance attributes like color, shape, and material. Each object in videos can be uniquely identified

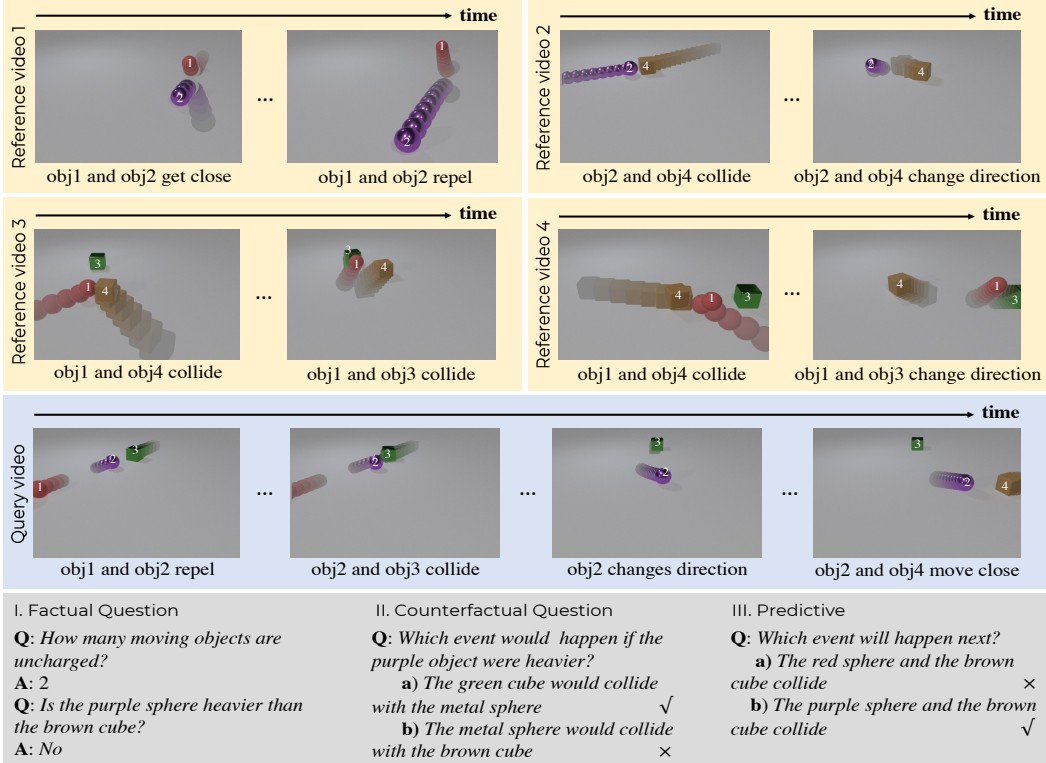

Figure 2: Sample target video, reference videos and question-answer pairs from ComPhy.

by these three attributes for simplicity. There are events, *in*, *out*, *collision*, *attraction* and *repulsion*. These object appearance attributes and events form the basic concepts of the questions in ComPhy.

**Physical Property.** Previous benchmarks (Riochet et al., 2018; Yi et al., 2020) mainly study appearance concepts like color and collision that can be perceived in even a single frame. In ComPhy, we additionally study *intrinsic* physical properties, *mass* and *charge*, which can not be directly captured from objects' static appearance. As shown in Fig. 1 (a), objects with same or opposite *charge* will repel or attract each other while objects without *charge* will not affect each other's motion without *collision*. As shown Fig. 1 (b), the object with larger *mass* (inertia) tends to maintain its original moving direction after the collision while the light object changes much more in its moving direction. Note that these *intrinsic* physical properties are orthogonal to the appearance attributes and can be combined with each other to generate more complicated and diverse dynamic scenes. For simplicity, ComPhy contains two mass values (*heavy* and *light*) and three charge types (*positive charged*, *negative charged* and *uncharged*). Theoretically, we can add more physical properties like *bounciness coefficients* and *friction* into ComPhy and make their values continuous. However, such a design will make the dataset too complicated and even difficult for people to infer the properties.

**Video Generations.** For each video, we first use a physical engine (Coumans & Bai, 2016–2021) to simulate objects' motions and then adopt a graphs engine (Community, 2018) to render frame sequences. Each target video for question answering contains 3 to 5 objects with random composition between their appearance attributes and physical properties. We set the length of the target video to be 5 seconds and additionally simulate the 6-th and 7-th seconds of the target video for predictive question annotation. We provide more video generation details in the Appendix.

**Task Setup.** It is not trivial to design a task setup to evaluate models' ability for physical reasoning since physical properties are not observable in a static frame. A straightforward design is to correlate object appearance with the physical property like "*red object is heavy*", "*yellow object is light*" and then ask "*what would happen if they collide*". However, such a setting is imperfect since it can not evaluate whether a model really understands the physical properties or just memorize the visual appearance prior. An ideal setting should be able to evaluate whether a model is like a human that can identify objects' properties from objects' motion and interactions with each other in the dynamic scenes and make the corresponding dynamic predictions.

To achieve this, we design a meta setting for physical reasoning, which provides few reference video samples along with the target video for models to infer objects' physical properties and then ask

questions about the objects' physical properties and dynamics. We show a sample of the dataset in Fig. 2. Each set contains a target video, 4 reference videos, and some questions about the visual attributes, physical properties, events, and dynamics of the target video. Objects in each set share the same visual attributes (color, shape, and material) and *intrinsic* physical property (mass and charge).

**Reference Videos.** To provide abundant visual content for physical property inference, we additionally provide 4 reference videos for each target video. We sample 2-3 objects from the target video, provide them with different initial velocities and locations, and make them interact (attract, repel or collide) with each other. The generation of the reference video follows the same standard as the target video, but the length of the videos is set to 2 seconds for scaling up. The interaction among objects in reference videos helps models to inference objects' properties. For example, the repulsion in Reference video 1 of Fig. 2 can help us identify that *object 1* and *object 2* carrying the same charge.

## 3.2 QUESTIONS

Following Johnson et al. (2017) and Yi et al. (2020), we develop a question engine to generate questions for factual, predictive, and counterfactual reasoning. Each question is paired with a functional program, which provides a series of explicit reasoning steps. We set all factual questions to be "open-ended" that can be answered by a single word or a short phrase. We set predictive questions and counterfactual questions to be multiple-choice and require models to independently predict each option is true or false. We provide question templates and examples in the Appendix.

**Factual.** Factual questions test models' ability to understand and reason about objects' physical properties, visual attributes, events, and their compositional relations. Besides the factual questions in existing benchmarks (Yi et al., 2020; Ates et al., 2020), as shown in the samples in Fig. 2 (I), ComPhy includes novel and challenging questions asking about objects' physical properties, charge and mass.

**Predictive.** Predictive questions evaluate models' ability to predict and reason about the events happening after the observed target video ends. It requires a model to observe objects' location and velocity at the end, identify objects' physical property and predict what will or not happen next.

**Counterfactual Charge and Mass.** Counterfactual questions ask what would happen on a certain hypothetical condition. ComPhy targets at reasoning the dynamics under the hypothesis that a certain object carrying a different physical property. Fig. 2 (II) shows typical question samples. Previous work (Yi et al., 2020; Riochet et al., 2018) also has counterfactual questions. However, they are merely based on the hypothesis that an object is removed rather than an object is carrying a different physical property value, which has different or even opposite effects on object motion prediction.

## 3.3 BALANCING AND STATISTICS

Overall, ComPhy has 8,000 sets for training, 2,000 sets for validation, and 2,000 for testing. It contains 41,933 factual questions, 50,405 counterfactual questions and 7,506 predictive questions, occupying 42%, 50% and 8% of the dataset, respectively. For simplicity, we manually make sure that a video set will only contain a pair of charged objects if it contains charged objects. Similarly, a video will only contain a heavy object or no heavy objects. A non-negligible issue of previous benchmarks like CLEVRER is its bias. As pointed out by Ding et al. (2020), about half of the counterfactual objects in CLEVRER have no collisions with other objects and the counterfactual questions can be solved merely based on the observed target videos. In ComPhy, we manually remove counterfactual questions on objects that have no interaction with other objects. When generating questions for comparing mass values and identifying charge relations between two objects, we systematically control that the two objects should have at least one interaction in one of the provided few video examples. We make sure that the few video examples are informative enough to answer questions based on the questions' programs and the video examples' property and interaction annotation.

## 4 EXPERIMENTS

In this section, We evaluate various baselines and analyze their results to study ComPhy thoroughly.

## 4.1 BASELINES

Following CLEVRER, We evaluate a range of baselines on ComPhy in Table 2. Such baselines can be divided into three groups, bias-analysis models, video-question answering models, and compositional reasoning models. To provide extensive comparison, we also implement some variant models that make use of both the target video and reference videos.

| Methods | Factual | Predictive | | Counterfactual | |
|---|---|---|---|---|---|
| | | per opt. | per ques. | per opt. | per ques. |
| Random | 29.7 | 51.9 | 22.6 | 49.7 | 9.1 |
| Frequent | 30.9 | 56.2 | 25.7 | 50.3 | 8.7 |
| Blind-LSTM | 39.0 | 57.9 | 28.7 | 55.7 | 12.5 |
| CNN-LSTM (Antol et al., 2015) | 46.6 | 59.5 | 29.8 | 58.6 | 14.6 |
| HCRN (Le et al., 2020) | 47.3 | 62.7 | 32.7 | 58.6 | 14.2 |
| MAC (Hudson & Manning, 2018) | **68.6** | 60.2 | 32.2 | 60.2 | 16.0 |
| ALOE (Ding et al., 2020) | 54.3 | 65.9 | 35.2 | 65.4 | 20.8 |
| CNN-LSTM (Ref) (Antol et al., 2015) | 41.9 | 59.6 | 29.4 | 57.2 | 12.8 |
| MAC (Ref) (Hudson & Manning, 2018) | 65.8 | 60.2 | 30.7 | 60.3 | 14.3 |
| ALOE (Ref) (Ding et al., 2020) | 57.7 | **67.9** | **37.1** | **67.9** | **22.2** |
| Human Performance | 90.6 | 88.0 | 75.9 | 80.0 | 52.9 |

Table 2: Evaluation of physical reasoning on ComPhy. Human performance is based on sampled questions. See Section 4.2 for more details.

**Baselines.** The first group of models are bias analysis models. They analyze the language bias in ComPhy and answer questions without visual input. Specifically, **Random** randomly selects an answer based on its question type. **Frequent** chooses the most frequent answer for each question type. **Blind-LSTM** uses an LSTM to encode the question and predict the answer without visual input. The second group of models are visual question answering Models. These models answers questions based on the input videos and questions. **CNN-LSTM** (Antol et al., 2015) is a basic question answering model. We use a resNet-50 to extract frame-level features and average them over the time dimension. We encode questions with the last hidden state from an LSTM. The visual features and the question embedding are concatenated to predict answers. **HCRN** (Le et al., 2020) is a popular model that hierarchically models visual and textual relationships. The third group of models are visual reasoning models. **MAC** (Hudson & Manning, 2018) decomposes visual question answering into multiple attention-based reasoning steps and predicts the answer based on the last step. **ALOE** (Ding et al., 2020) achieves state-of-the-art performance on CLEVRER with transformers (Vaswani et al., 2017).

**Baselines with Reference Videos.** We implement some variants of existing baseline models, which adopt both the target video and reference videos as input. We develop **CNN-LSTM (Ref)**, **MAC (Ref)** and **ALOE (Ref)** from **CNN-LSTM**, **MAC** and **ALOE** by concatenating the features of both reference videos and the target video as visual input.

**Evaluation.** We use the standard accuracy metric to evaluate the performance of different methods. For multiple-choice questions, we report both the per-option accuracy and per-question accuracy. We consider a question is correct if the model answers all the options of the question correctly.

## 4.2 EVALUATION ON PHYSICAL REASONING.

We summarize the question-answering results of different baseline models in Table 2. We also find that models have different relative performances on different kinds of questions, indicating that different kinds of questions in ComPhy require different reasoning skills.

**Factual Reasoning.** Factual questions in ComPhy require models to recognize objects' visual attributes, analyze their moving trajectories and identify their physics property to answer the questions. Based on the result, we have the following observation. First, we find that the "blind" models, **Random**, **Frequent** and **Blind-LSTM** achieve much worse performance than other video question answering models and reasoning models. This shows the importance of modeling both visual context and linguistic information on ComPhy. We also find that video question answering models **CNN-LSTM** and **HCRN** perform worse than visual reasoning models **MAC** and **ALOE**. We believe the reasons are that models like **HCRN** are mostly designed for human action video, which mainly focuses on temporal modeling of action sequences rather than spatial-temporal modeling of physical events. Among all the baselines, we find that **MAC** performs the best on factual questions, showing the effectiveness of its compositional attention mechanism and iterative reasoning processes.

**Dynamcis Reasoning.** An important feature of ComPhy is that it requires models to make counterfactual and future dynamic predictions based on their identified physical property to answer the questions. Among all the baselines, we find that **ALOE (Ref)** achieves the best performance on counterfactual and future reasoning. We think the reason is that the self-attention mechanism and object masking self-supervised techniques provides the model with the ability to model spatio-temporal visual context and imaging counterfactual scenes to answer the questions.

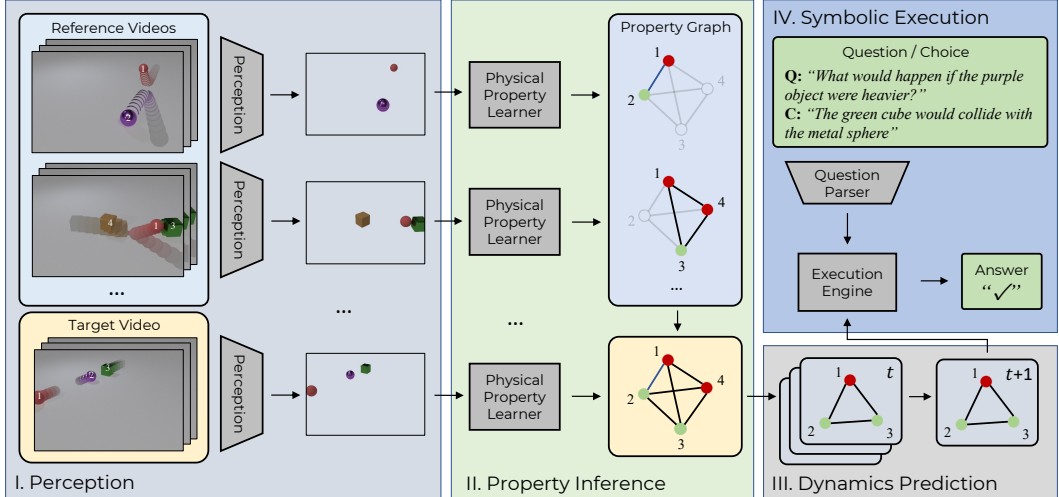

Figure 3: The perception module detects objects' location and visual appearance attributes. The physical property learner learns objects' properties based on detected object trajectories. The dynamic predictor predicts objects' dynamics in the counterfactual scene based on objects' properties and locations. Finally, an execution engine runs the program parsed by the language parser on the predicted dynamic scene to answer the question.

**Reasoning with Reference Videos.** We observe that **CNN-LSTM (Ref)** and **MAC (Ref)** achieve only comparable or even slightly worse performance comparing with their original models, **CNN-LSTM** and **MAC**. We also find that **ALOE (Ref)** achieves better performance than **ALOE**. However, the increment from **ALOE** to **ALOE (Ref)** is limited. All these variant models can not get much gain from their original models by concatenating the reference videos as additional visual input. We think the reason is that these models are based on massive training videos and question-answer pairs and have difficulties adapting to the new scenario in ComPhy, which requires models to learn the new compositional visible and hidden physical properties from only a few examples.

**Human Performance.** To assess human performance on ComPhy, 14 people participated in the study. We require all the participants to have basic physics knowledge and can speak fluent English. Participants were first shown with a few demo videos and questions to test that they understood the visual events, physical properties, and the text description. Participants are then asked to answer 25 question samples of different kinds in ComPhy. And we record accuracy of $90.6\%$ for factual questions, $88.0\%$ for predictive per option, $80.0\%$ counterfactual per option, $75.9\%$ for predictive per question and $52.9\%$ for counterfactual per question. This shows that while our questions are difficult for machine models, humans can still well understand the physical properties and make dynamic predictions in ComPhy to answer the corresponding questions.

## 5 COMPOSITIONAL PHYSICS LEARNER

### 5.1 MODEL

We propose an oracle model that performs compositional dynamics reasoning based on a neural-symbolic framework. Inspired by the recent models (Yi et al., 2018; 2020), we factorize the process of physical reasoning into several stages. As shown in Fig. 3, our model consists of four major stages: perception (I), property inference (II), dynamic prediction (III), and symbolic dynamics reasoning (IV). Given a target video, 4 reference videos, and a query question, we first detect objects' location and appearance attributes with a perception module. Objects' trajectories are fed into a physical property learner to learn their physical properties. Given objects' properties and location, the dynamic predictor predicts objects' dynamics based on the physical properties. Finally, an execution engine runs the program from the language parser on the predicted objects' motion to answer the question.

Our model's core contribution lies in a physical property learner that infers the hidden per-object and pairwise properties from the videos and a dynamics model that predicts dynamics based on the inferred physical properties. These modules enable our model to tackle challenging physics reasoning tasks that involve physical properties that are not directly observable.

**Perception.** Our perception module detects and recognizes all objects at every video frame. Given an input frame, our model first applies a Mask-RCNN (He et al., 2017) to detect all objects in the frame and extract corresponding visual attributes like color, shape, and material. The perception module outputs the static attributes of all objects as well as their full-motion trajectories during the video. Please refer to (Yi et al., 2018) for further details.

**Physical Property Inference.** The physical property learner (PPL) lies at the core of our model's capability to tackle complex and compositional physical interactions. Given the object motion trajectories from all reference and target videos, the PPL predicts the mass and relative charge between each object pair in the video. Under the hood, PPL is implemented using a graph neural network (Kipf et al., 2018) where the node features contain object-centric property (mass) and edge features encode pairwise properties (relative charge).. Given the input trajectories $\{\mathbf{x}_i\}_{i=1}^N$ of $N$ objects in the video, PPL performs the following message passing operations,

$$\mathbf{v}_i^0 = f_{emb}(\mathbf{x}_i), \quad \mathbf{e}_{i,j}^l = f_{rel}^l(\mathbf{v}_i^l, \mathbf{v}_j^l), \quad \mathbf{v}_i^{l+1} = f_{enc}^l(\sum_{i \neq j} \mathbf{e}_{i,j}^l), \tag{1}$$

where $l \in [0, 1]$ denotes the message passing steps and $\mathbf{x}_i$ denotes the concatenation of the normalized detected object coordinates $\{x_{i,t}, y_{i,t}\}_{t=1}^T$ over all $T$ frames. $f_{(\dots)}$ are functions implemented by fully-connected layers. We then use two fully-connected layer to predict the output mass label $f_v^{pred}(\mathbf{v}_i^2)$ and edge charge label $f_e^{pred}(\mathbf{e}_{i,j}^1)$, respectively.

We further note that because the system is invariant under charge inversion, the charge property is described by the relative signs between each object pair, even though the charge carried by each individual object should be easy to recover given the pairwise relation plus the true sign of one object. The full physical property of a video set can be represented as a fully connected property graph. Each node represents an object that appears in at least one of the videos from the set. And each edge represents if the two nodes the edge connects carry the same, opposite, or no relative charge (that is, one or both objects are charge-neutral). As shown in Fig. 3(II), for each reference video, PPL only predicts the properties of the objects it covers, revealing only parts of the property graph. We align the predictions of different nodes and edges by objects' static attributes predicted by the perception module. The full object properties are obtained by combining all the subgraphs generated by each video from the set via max-pooling over node and edge.

**Dynamic Prediction based on Physical Property.** Given the object trajectories at the $t$-th frame and their corresponding object properties (i.e., mass and charge), we need to predict objects' positions at the $t + 1$ frame. We achieve this using a graph-neural-network-based dynamic predictor. We represent the $i$-th object at the $t$-th frame with $\mathbf{o}_i^{t,0} = ||_{t-3}^t (x_i^t, y_i^t, w_i^t, , h_i^t, m_i)$, which is a concatenation of the object location $(x_i^t, y_i^t)$, size $(w_i^t, h_i^t)$ and the mass label $(m_i)$ over a history window of 3. We present objects by the locations of a history rather than the only location at the $t$-th frame to encode the velocity and account for the perception error. Specifically, we have

$$\mathbf{h}_{i,j}^{t,0} = \sum_k z_{i,j,k} g_{emb}^k(\mathbf{o}_i^{t,0}, \mathbf{o}_j^{t,0}), \quad \mathbf{o}_j^{t,l+1} = \mathbf{o}_j^{t,l} + g_{rel}^l(\sum_{i \neq j} (\mathbf{h}_{i,j}^{t,l})),$$
$$\mathbf{h}_{i,j}^{t,l+1} = \sum_k z_{i,j,k} g_{enc}^{k,l}([\mathbf{o}_i^{t,l+1}, \mathbf{o}_i^{t,0}], [\mathbf{o}_j^{t,l+1}, \mathbf{o}_j^{t,0}]), \tag{2}$$

where $k \in \{0, 1, 2\}$ represents if the two connected nodes carry the same, opposite, or no relative charge. $z_{i,j,k}$ are the $k$-th element of the one-hot indication vector $\mathbf{z}_{i,j}$. $l \in [0, 1]$ denotes the message passing steps and, $g_{(\dots)}$ are functions implemented by fully-connected layers. We predict object location and size at the $t + 1$-th frame using a function that consists of one fully connected layer $g_{pred}(\mathbf{o}_j^{t,2})$. Given the target video, we predict the future by initializing dynamic predictor input with the last three frames of the target video and then iteratively predict the future by feeding the prediction back to our model. We get the physical property-based prediction for the counterfactual scenes by using the first 3 frames in the target video as input and updating their mass label $(m_i)$ and one-hot indicator vector $\mathbf{z}_{i,j}$ correspondingly.

**Symbolic Execution.** With the object visual attributes, physical property, and dynamics ready, we use a question parser to transform the question and choices into functional programs and perform step-by-step symbolic execution to get the answer. We provide more details in the Appendix.

## 5.2 PERFORMANCE ANALYSIS

**Effectiveness of Physical Property Learning.** We report the performance of the proposed oracle CPL in Table 3. We can see that CPL shows better performance on all kinds of questions compared

| Methods | Factual | Predictive | | Counterfactual | |
|---|---|---|---|---|---|
| | | per opt. | per ques. | per opt. | per ques. |
| CPL | 80.5 | 75.3 | 56.4 | 68.3 | 29.1 |
| NS-DR+ | - | 73.3 | 50.8 | 61.1 | 16.6 |
| CPL-Gt | 100 | 87.6 | 74.0 | 74.0 | 35.3 |

Table 3: Evaluation of CPL on ComPhy.

Figure 4: Generalization of physical reasoning.

to the baseline methods in Table 2. The high accuracy on factual questions shows that CPL is able to inference objects' physical properties from few video examples and combine them with other appearance properties and events for question answering. Specifically, we compare the mass and charge edge label prediction result with the ground-truth labels and find that they achieve an accuracy of 90.4% and 90.8%. This shows that that the PPL in CPL are able to learn physical properties from objects' trajectories and interactions in a supervised manner. We also implement a graph neural network baseline that only relies on the target video without reference videos. It achieves an accuracy of 59.4% and 70.2% on mass and charge edge labels. This indicates the importance of reference videos for physical property inference. For better analysis, we have reported a baseline with ground-truth object properties and visual static and dynamic attributes (CPT-Gt). Although it achieves constant gains, there is still much room for further improvement. This shows the bottleneck of ComPhy lies in predicting physical-based dynamics.

**Dynamics Reasoning.** The better performance in counterfactual and predictive questions indicates that CPL can image objects' motions in counterfactual and future scenes based on the identified physical properties to some extent. We also notice that there is still a large gap between CPL's performance and human performance shown in Section 4.2 especially on counterfactual reasoning. We find that the dynamic predictor in CPL still shows its limitation on long-term dynamic prediction. This suggests that a stronger dynamic predictor may further boost CPL's performance.

**Comparison between CPL and NS-DR+.** We also implement a variant of the NS-DR (Yi et al., 2020) model for better analysis. NS-DR does not explicitly consider changes in physical properties like mass and charge; thus it cannot run the symbolic programs related to the physical properties in ComPhy. To run NS-DR successfully in ComPhy, we provide NS-DR with extra ground-truth physical property labels. The variant, NS-DR+ uses the PropNet (Li et al., 2019b) for dynamics predictions, which does not consider the mass and charge information of the objects. Compared to our model, NS-DR+ assumes extra information and can directly answer factual questions using the supplied ground-truth labels; thus we ignore the comparison on factual questions and focus on the evaluation on the models' ability for dynamic prediction. As shown in Table 3, CPL achieves much better performance than NS-DR+ on counterfactual questions and predictive questions, showing the importance of modeling mass and charges on nodes and edges of the graph neural networks.

**Generalization to More Complex Scenes.** To test the model's generalization ability, we additionally generate a new dataset that contains videos with more objects (6 to 8) and contains both an object with a large mass and a pair of charged objects. We show the performance of the best baseline models **ALOE**, **ALOE (Ref)** and CPL in Fig. 4. Both **ALOE** models and CPL have a large drop in question-answering performance. We believe the reason is that the transformer layer in **ALOE**, the graph neural network based-physical property learner and dynamics predictor in CPL show limitation on more complex visual dynamic scenes, which has a different statistic distribution as the scenes in the original training set. We leave the development of more powerful models with stronger generalization ability as future work.

## 6 CONCLUSIONS

In this paper, we present a benchmark named ComPhy for physical reasoning in videos. Given few video examples, ComPhy requires models to identify objects' intrinsic physical properties and predict their dynamics in counterfactual and future scenes based on the properties to answer the questions. We evaluate various existing state-of-the-art models on ComPhy, but none of them achieves satisfactory performance, showing that physical reasoning in videos is still a challenging problem and needs further exploration. We also develop an oracle model named CPL, using graph neural networks to infer physical properties and predict dynamics. We hope that ComPhy and CPL can attract more research attention in physical reasoning and building more powerful physical reasoning models.

**Acknowledgement** This work was supported by MIT-IBM Watson AI Lab and its member company Nexplore, ONR MURI, DARPA Machine Common Sense program, ONR (N00014-18-1-2847), and Mitsubishi Electric.

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

## A  EXAMPLES FROM COMPHY

Here we provide more examples from ComPhy in Fig. 5 and Fig. 6. From these examples, we can see the following features of ComPhy. First, to answer the factual questions, models not only need to recognize objects' visual appearance attributes and events in the video but also identify their intrinsic physical properties from the given video set. Second, to answer counterfactual and predictive questions, models needs to predict objects' dynamics in counterfactual or future scenes, which can be severely affected by intrinsic physical properties. We also show some typical question and choice samples as well as their underlying reasoning program logic in Fig. 7 and Fig. 8.

## B  VIDEO GENERATION

We provide more details for video generation. The generation of the videos in ComPhy can be decomposed into two steps. First, we adopt a physical engine Bullet Coumans & Bai (2016–2021) to simulate objects' motions and their interactions with each other. Since Bullet does not officially support the effect of electronic charges, we add external forces between charged objects, whose values are inversely proportional to the square of the objects' distance, to simulated Coulomb forces. We assign the *light* object with a mass value of 1 and assign *heavy* object with a mass value of 5. We manually make sure that each reference video at least contain an interaction (collision, charge and mass) among objects to provide enough information for physical property inference. And each object should appear at least once in the reference videos. The simulated objects' motions are sent to BlenderCommunity (2018) to render high-quality image sequences.

## C  QUESTION TEMPLATES

We show the new question templates that have not been introduced before in table 4. As shown in the table, we can see that the new question templates have more symbolic operators related to the physical properties. Phrases like "heavy moving spheres" and "charged cubes" require models to infer objects' physical property values; phrases like "... heavier than ..." require models to compare the relative physical property values of two objects. For counterfactual questions, we have new conditions like "If the cyan object were uncharged" and "If the sphere were lighter". They aim to reason the dynamics under the hypothesis that a certain object carrying a different physical property. Such new features in language make ComPhy unique and challenging.

## D  CPL DETAILS

In this section, we provide more details for the proposed Compositional Physics Learner (CPL). Inspired by the NS-DR model in (Yi et al., 2020), CPL decomposes physical reasoning in ComPhy into four main components, visual perception, physical property learning, property based dynamic predictor and symbolic execution.

The symbolic execution component first adopts a program parser to parse the query question into a functional program, containing a series of neural operations. The program parser is an attention-based seq2seq model Bahdanau et al. (2015), whose input is the word sequence in the question/choice and output is the sequence of neural operations. The symbolic executor then executes the operations on the predicted dynamic scene to get the answer to the question. We summarize all the symbolic operations in CPL in table 5. Compared with the previous benchmarks Yi et al. (2020); Ates et al. (2020), ComPhy has more operation on physical property identification, comparison and corresponding dynamic prediction.

We train the Mask-RCNN He et al. (2017) in the perception module with 4,000 frames randomly-selected from the training set of ComPhy. We train the program parser and the property concept learner with program and property labels using cross-entropy loss. We optimize the dynamic predictor with mean square error loss between the predicted objects' trajectories and the detected objects' trajectories by the perception module. We train the all the modules using Pytorch library Paszke et al. (2017) on Titan Nvidia GTX 1080-Ti GPUs.

| Question Type | Template and Example |
|---|---|
| Counterfact mass | If the _SA_ were _MP_, _Q_?
If the sphere were lighter, which event would not happen? |
| Counterfact charge | If the _SA_ were _CP_, _Q_?
If the cyan object were uncharged, which event would happen? |
| Query | What is the _H_ of the _DA_ _SA_ that is _PA_?
What is the color of the moving cylinder that is heavy? |
| Exist | Are there any _PA_ _DA_ _SA_ _TI_?
Are there any charged stationary cube? |
| Count | How many _PA_ _DA_ _SA_ are there _TI_?
How many heavy moving spheres are there when the video ends? |
| Mass compare | Is the _DA1_ _SA1_ heavier than the _DA2_ _SA2_?
Is the blue sphere heavier than the gray cube? |
| Compare charge 1 | Are the _DA1_ _SA1_ and the _DA2_ _SA2_ oppositely charged?
Are the blue sphere and the purple sphere oppositely charged? |
| Compare charge 2 | Are the _DA1_ _SA1_ and the _DA2_ _SA2_ with the same type of charge?
Are the blue cube and the brown cylinder with the same type of charge? |
| Query both | What are the _Hs_ of the two objects that are charged?
What are the colors of the two objects that are charged? |

Table 4: Question templates and examples in ComPhy. _SA_ denotes static attributes like "red" and "sphere"; _DA_ denotes dynamic attributes, "moving" and "stationary"; _MP_ denotes mass attributes, "lighter" and "heavier"; _Q_ denotes question phrases like "which of the following would happen"; _CP_ denotes charge attributes, "uncharged" and "oppositely charged"; _H_ denotes visible concepts, "color", "shape" and material; _PA_ denotes physical attributes, heavy, light, charged and "uncharged"; _TI_ denotes time indicators like "when the video ends".

## E  BASELINE IMPLEMENTATION DETAILS

In this section, we provide more details for baselines in the experimental section. We implement baselines based on the publicly-available source code. For multiple-choice questions, we independently concatenate the words of each option and the question as a binary classification question. Similar to CLEVRER (Yi et al., 2020), we use ResNet-50 (He et al., 2016) to extract visual feature sequences for **CNN+LSTM** and **MAC** and variants with reference videos. We evenly sample 25 frames for each target video and 10 frames for each reference video. For **HCRN**, we use appearance feature from ResNet-101 (He et al., 2016) and motion feature from ResNetXt-101 (Xie et al., 2017; Hara et al., 2018) following the official implementation. For **ALOE**, we use MONet(Burgess et al., 2019) to extract visual representation and sample 25 frames for each target video. For **ALOE (Ref)**, we sample 10 frames for each reference video and concatenate the reference frames and the target frames as visual representation. We train all the models until they are fully converged, select the best checkpoint on the validation set and finally test on the testing set.

| Type | Operation | Signature |
|---|---|---|
| Counterfact Operation | `Counterfactual_mass_heavy`
Return all events after making the object heavy
`Counterfactual_mass_light`
Return all events after making the object light
`Counterfactual_uncharged`
Return all events after making the object uncharged
`Counterfactual_opposite_charged`
Return all events after making the object oppositely charged | $(object) \rightarrow events$

$(object) \rightarrow events$

$(object) \rightarrow events$

$(object) \rightarrow events$ |
| Object Property Operations | `filter_heavy`
select all the heavy objects
`filter_light`
select all the light objects
`filter_charged`
select all the charged objects
`filter_uncharged`
select all the uncharged objects | $(objects) \rightarrow objects$

$(objects) \rightarrow objects$

$(objects) \rightarrow objects$

$(objects) \rightarrow objects$ |
| Object Appearance Operations | `Filter_static_attr`
Select objects from the input list with the input static attribute
`Filter_dynamic_attr`
Selects objects in the input frame with the dynamic attribute | $(objects, attr) \rightarrow objects$

$(objects, attr, frame) \rightarrow objects$ |
| Event Operations | `Filter_event`
Select all events that involve the input objects
`Get_col_partner`
Return the collision partner of the input object
`Filter_before`
Select all events before the target event
`Filter_after`
Select all events after the target event
`Filter_order`
Select the event at the specific time order
`Get_frame`
Return the frame of the input event in the video | $(events, objects) \rightarrow events$

$(event, object) \rightarrow object$

$(events, events) \rightarrow events$

$(events, events) \rightarrow events$

$(events, order) \rightarrow event$

$(event) \rightarrow frame$ |
| Others | `Unique`
Return the only event/object in the input list | $(events/objects) \rightarrow event/object$ |
| Input Operations | `Start`
Returns the special "start" event
`end`
Returns the special "end" event
`Objects`
Returns all objects in the video
`Events`
Returns all events happening in the video
`UnseenEvents`
Returns all future events happening in the video | $() \rightarrow event$

$() \rightarrow event$

$() \rightarrow objects$

$() \rightarrow events$

$() \rightarrow events$ |
| Output Operations | `Query_both_attribute`
Returns the attributes of the input two objects
`Query_direction`
Returns the direction of the object at the input frame
`Is_heavier`
Returns "yes" if *obj1* is heavier than *obj2*
`Is_lighter`
Returns "yes" if *obj1* is lighter than *obj2*
`Query_attribute`
Returns the attribute of the input objects like color
`Count`
Returns the number of the input objects/ events
`Exist`
Returns "yes" if the input objects is not empty
`Belong_to`
Returns "yes" if the input event belongs to the input event sets
`Negate`
Returns the negation of the input boolean | $(object, object) \rightarrow attr$

$(object, frame) \rightarrow attr$

$(obj1, obj2) \rightarrow bool$

$(obj1, obj2) \rightarrow bool$

$(object) \rightarrow attr$

$(objects) \rightarrow int$
$(events) \rightarrow int$
$(objects) \rightarrow bool$

$(event, events) \rightarrow bool$

$(bool) \rightarrow bool$ |

Table 5: Symbolic operations of CPL on ComPhy. In this table, "order" denotes the chronological order of an event, e.g. "First" and "Last"; "static attribute" denotes object static concepts like "Red" and "Rubber" and "dynamic attribute" represents object dynamic concepts like "Moving".

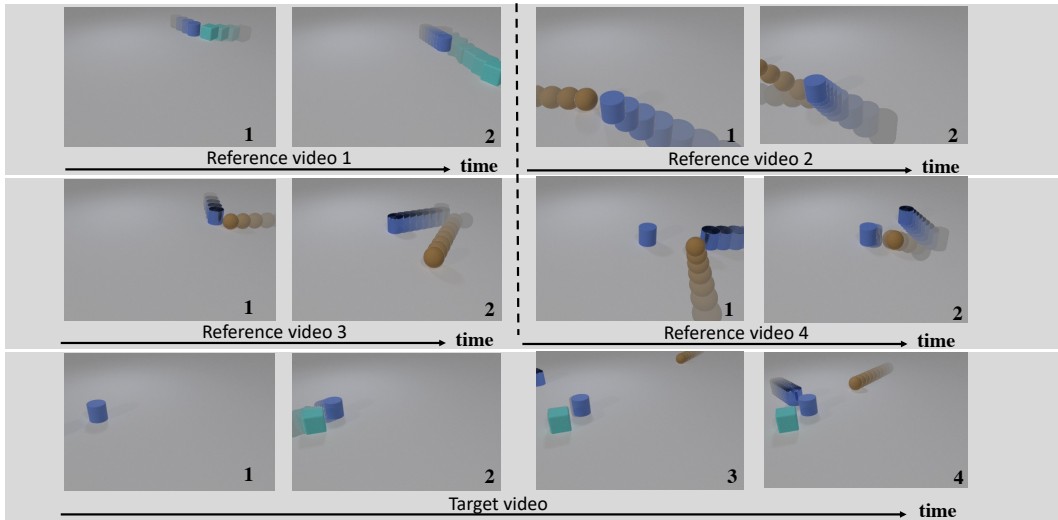

### I. Factual
**Q1**: Is the cyan cube heavier than the rubber cylinder? A: No.
**Q2**: Are there any blue cylinders that enter the scene?  **A**:Yes.

### II. Counterfactual
**Q3**: If the rubber cylinder were lighter, which of the following would happen?
    **a**) The cube would collide with the rubber cylinder  √
    **b**) The rubber cylinder and the sphere would collide  √
    **c**) The metal object would collide with the sphere  ×

### III. Predictive
**Q4**: What will happen next?
    **a**) The rubber cylinder and the metal object collide √
    **b**) The rubber cylinder and the sphere collide  √
    **c**) The cube collides with the sphere ×

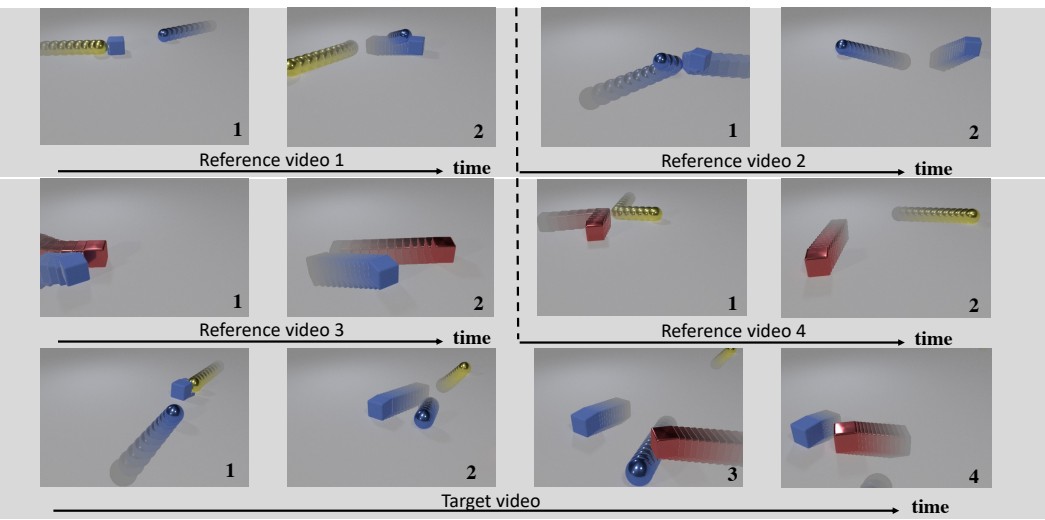

### I. Factual
**Q1**: What are the colors of the two objects that are charged?  **A1**: Yellow and blue.
**Q2**: Are there any metal cubes that enter the scene?  **A2**: No.
**Q3**: What is the direction of the blue cube when the video ends? **A3**: Left.

### II. Counterfactual
**Q3**: If the blue sphere were oppositely charged, what would happen?
    **a**) The yellow sphere and the rubber cube would collide √
    **b**) The yellow object and the blue sphere would collide √
    **c**) The blue cube and the metal cube would collide ×
    **d**) The yellow object and the red object would collide ×

### III. Predictive
**Q4**: Which event will happen next?
    **a**) The blue cube and the red cube collide √
    **b**) The blue sphere collides with the metal cube  ×

Figure 5: Sample target video, reference videos and question-answer pairs from ComPhy.

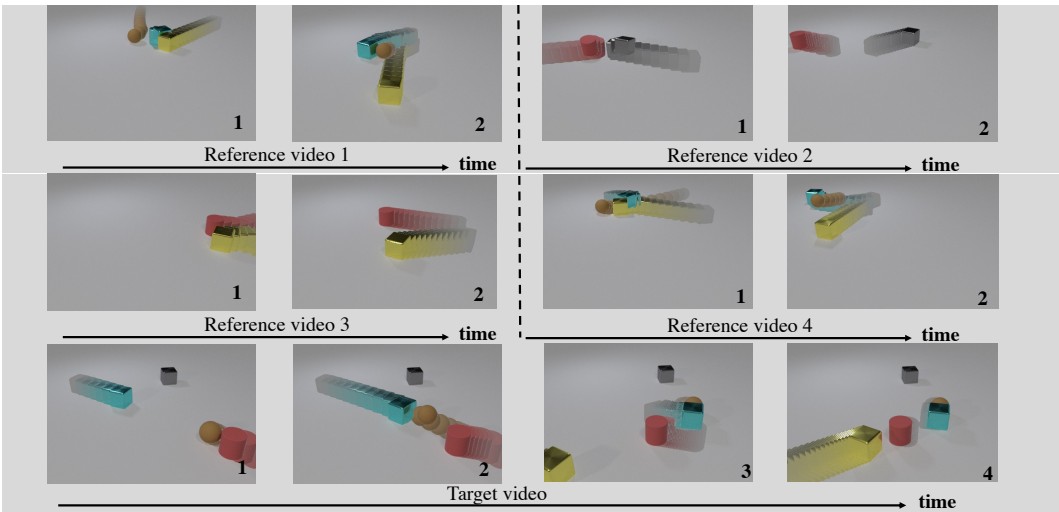

**I. Factual**

**Q1**: What color is the moving rubber object that is uncharged? **A1**: Rubber.

**Q2**: In which direction is the red cylinder moving when the yellow object enters the scene? **A2**: Left.

**Q3**: How many moving cyan objects are charged? **A3**: 1.

**II. Counterfactual**

**Q3**: If the cyan object were uncharged, which event would happen?

    **a)** The sphere and the cyan object would collide √

    **b)** The cylinder would collide with the sphere √

    **c)** The cylinder would collide with the gray cube ×

    **d)** The sphere would collide with the yellow object ×

**Q4**: If the sphere were oppositely charged, which of the following would happen?

    **a)** The cylinder and the sphere would collide √

    **b)** The cylinder would collide with the cyan cube ×

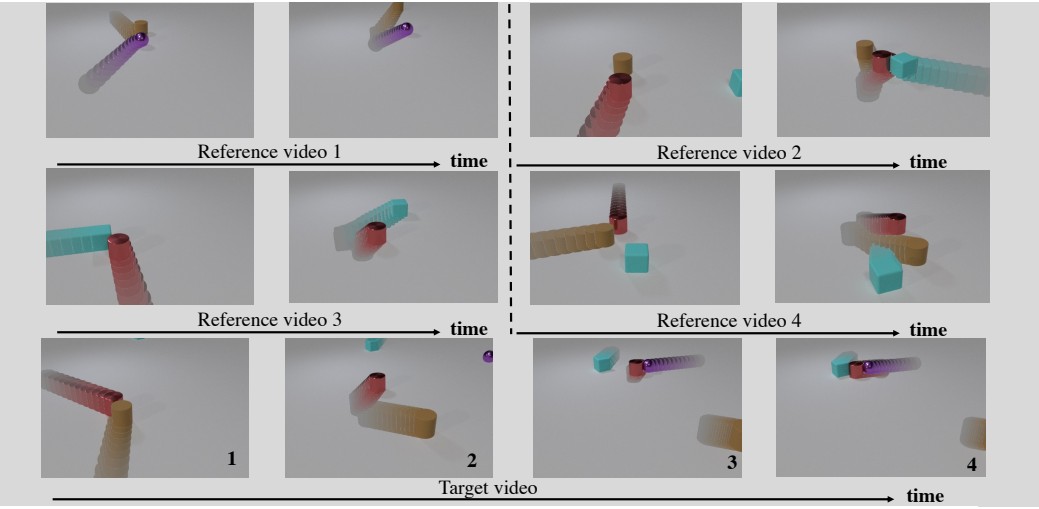

**I. Factual**

**Q1**: Is the moving purple object lighter than the moving brown object? **A1**: No.

**Q2**: What is the color of the moving metal object that is light? **A2**: Red.

**Q3**: Are there any brown cylinders that exit the scene after the sphere enters the scene? **A3**: Yes.

**II. Counterfactual**

**Q3**: If the sphere were lighter, which event would **not** happen?

    **a)** The red cylinder and the cube would collide √

    **b)** The sphere and the cube would collide √

    **c)** The red cylinder and the sphere would collide ×

    **d)** The metal cylinder and the rubber cylinder would collide×

**III. Predictive**

**Q4**: Which event will happen next?

    **a)** The sphere collides with the cube √

    **b)** The metal cylinder and the brown cylinder collide ×

Figure 6: Sample target video, reference videos and question-answer pairs from ComPhy.

**Q1**: How many heavy stationary objects are there when the video begins?

**Q2**: What color is the moving rubber object that is uncharged?

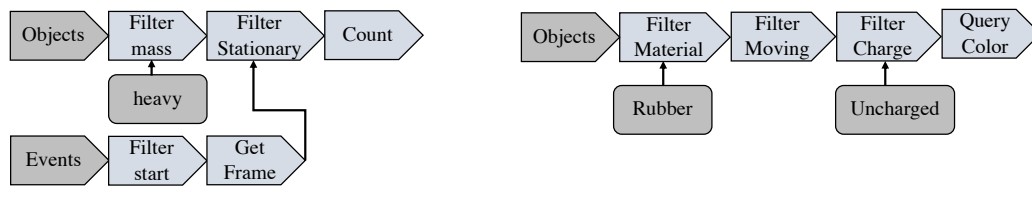

**Q3**: How many moving green objects are charged?

**Q4**: What shape is the moving metal object that is light?

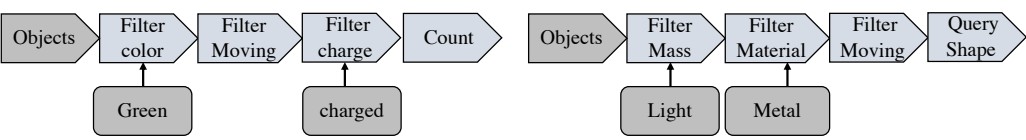

Figure 7: Sample of factual questions and their underlying functional programs in ComPhy.

**Q1**: If the rubber cylinder were heavier, which of the following would happen?

**Q2**: Which of the following would not happen if the sphere were uncharged?

**C1**: The cylinder and the cube would collide

**C2**: The blue object and the metal object would collide

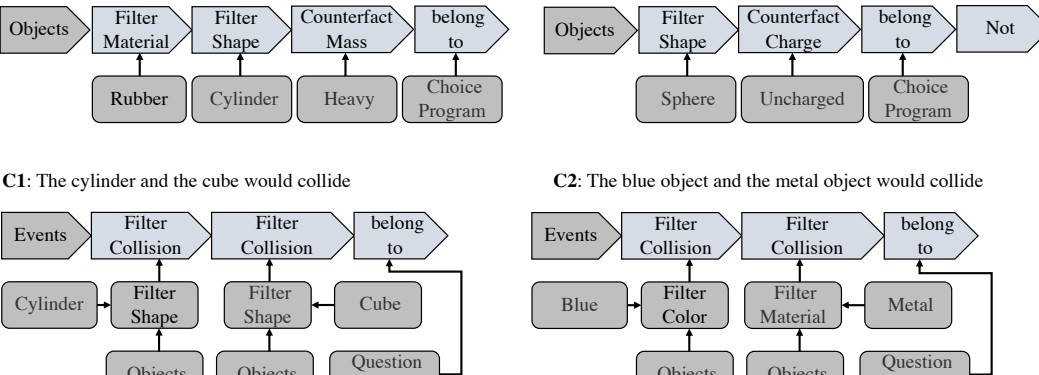

Figure 8: Sample of counterfactual questions, choice options and their underlying functional programs in ComPhy.

