# OpenReview forum: "ComPhy: Compositional Physical Reasoning of Objects and Events from Videos"
_ICLR.cc/2022/Conference — ICLR 2022 Poster_

### Official Review · Reviewer_DJEq · 2021-10-20

**Correctness:** 3
**Technical Novelty And Significance:** 2
**Empirical Novelty And Significance:** 2
**Recommendation:** 5
**Confidence:** 4

**Main Review:**

**Strengths**

- The paper explores a quite interesting direction. Reasoning about the physical properties of objects is a critical requirement for designing agents that interact with the world around them.

- The proposed CPL approach outperforms a number of baseline approaches.

**Weaknesses**

The paper has some shortcomings:

(1) The work is incremental compared to Yi et al., 2020. The only difference is that this paper infers hidden properties instead of collisions. The questions and the way they are generated are very similar as well.

(2) The dataset is not comprehensive enough. Only two properties are considered: mass and electricity charge. I don't think electricity charge is a useful property to infer since it can be used only in a limited set of applications. The dataset should include useful properties such as friction and material. Moreover, the synthetic videos are so simplistic (cubes and spheres moving on a plane). It is not clear if a system that solves this task can be useful in any scenario that is slightly more complex in terms of appearance and object interactions. I encourage the authors to create more realistic scenarios in simulators such as AI2-THOR and iGibson.

(3) It is not clear how the train/val/test splits are created. It doesn't seem there is much variability in the videos so there might be a high degree of similarity between train and test scenarios. Some metrics should be reported to show train and test splits are different enough.

**Questions and comments**

- "CNN-LSTM (Antol et al., 2015) is a basic video question-answering model.": This paper is not a video question-answering model. It just uses single frames.

- "We think the reason is that these models are based on massive training videos and question-answer pairs and have difficulties adapting to the new scenario in ComPhy": It might be simply due to the large networks and small amount of data.

- There is an assumption that all objects are already known and (it seems) there is only one object per each category. It would be nice to see the results of generalization to unseen shapes. However, it is not clear how the graph can accommodate new objects. It seems it is a fixed graph.

- Equation 2 is not clear. What is g_emb and g_enc? What does z represent?

- It is not clear why CPL shows better performance on counterfactual questions. There is no component in the model to better handle that type of question.

- Why is CPL called oracle?

**Summary Of The Paper:**

The paper focuses on inferring two hidden physical properties (mass and electricity charge). A synthetic dataset, called ComPhy is created. The dataset consists of simple shapes (such as cubes and sphere) that move on a flat plane. For each query video, there are four reference videos that the models can use to learn the properties of objects. The dataset also includes three types of questions: factual, counterfactual and predictive. The paper also presents a pipeline approach called CPL that leverages graphs to model the relationship of objects and solve the QA tasks.

**Summary Of The Review:**

The reasons for my low rating are: (1) lack of novelty with respect to previous works, (2) dataset not being comprehensive enough, (3) simplicity of the dataset and lack of generalization to slightly richer scenarios in terms of appearance and object variability (4) lack of clarity on creation of the dataset.

---

> ### Author Response · Authors · 2021-11-23
> **Response to Reviewer DJEq (Part 1)**
>
> Thank you for your constructive comments.
>
> **1. About ComPhy's Novelty**
>
> Please refer to the general response **1. ComPhy V.S. CLEVRER** to see the major differences between ComPhy and CLEVRER and the novelty of ComPhy.
>
> **2. About ComPhy's Complexity and More Physical Properties**
>
> Please refer to the general response **2. About ComPhy’s Complexity and More Physical Properties**.
>
> **3. About the difference between training/ validation/ test splits**
>
> **[Generation of Different Splits]**. As shown in Section 3.1, we generate the training/ validation/ test splits from the same distribution. Specifically, we sample each target video for question answering containing 3 to 5 objects with random composition between their appearance attributes and physical properties. We then sample 2-3 objects from the target video, provide them with different initial velocities and locations, and make them interact (attract, repel and collide) with each other to generate the videos.
>
> **[About the concern of overfitting and similarity between training and testing]**. From the experimental results, we can find that current models are still far from overfitting the test split that has a similar distribution to the training set. All current models achieve relatively low performance. Moreover, as shown in the paragraph of "Generalization to More Complex Scenes" in Section 5.2, we do have a generalization test split to test models' abilities to handle difficult dynamic scenes with more and diverse interacting objects that are different from the training split.
>
> **4. About the details of CNN-LSTM**
>
> Thanks for your reminder that CNN-LSTM(Antol et al., 2015) is an image question answering model. The performance shown in table 2 is a variant of CNN-LSTM(Antol et al., 2015), in which we first use a resNet-50 to extract frame-level features and average the feature sequence over the time dimension before fusing the textual and visual features. We will show more details in the revised version.
>
> **5. About the reason of baseline models' performance on ComPhy**
>
> We agree that large models and a small amount of data may be the key reason what existing models fail. And one specific feature of ComPhy is that it requires models to learn the new composition between physical property and visual attributes from a small amount of data.
>
> **6. Generalization to unseen objects**
>
> **About object assumption**. As described in Section 3.1 "Objects and Events", each object in videos can be uniquely identified by the three static attributes (color, shape, and material). There may be different numbers of objects in the target videos and reference videos.
>
> **About Handling New Objects**. As described in Section 5.1 "Physical Property Inference", the graph neural networks in CPL first predict each objects' physical properties independently. CPL does not assume objects are known or fixed during training and inference and it can handle videos of any number of objects. In CPL, we align objects in reference videos and the target video by objects' predicted static attributes. We get the final node and edge predictions by aligning nodes with their predicted static attributes from the perception module and max-pooling the predictions in all videos.
>
> **7. About the details of Equation 2**
>
> As described in the text below Equation 2, **g_emb** and **g_enc** (**g_(...)**) are functions implemented by fully-connected layers; **z_{i,j}** is a one-hot indicator vector indicating the predicted relation type between the **i**-th and the **j**-th nodes. **z_{i, j, k}** are the *k*-th element of the one-hot indicator vector, where **k in {0, 1, 2}** represents if the two connected nodes carry the same, opposite, or no relative charge.
>
> **8. About the reason why CPL is effective in counterfactual questions**
>
> As shown in the Paragraph "Dynamic Prediction based on Physical Property" of section 5.1, CPL has an explicit physical-property-based dynamic predictor for counterfactual situations, which is the main reason for CPL's effectiveness in counterfactual questions. The physical-based dynamic predictor in CPL encodes different types of relations (e.g. attraction, repulsion) with different edge types and different mass values with different object embeddings. In contrast, the PropNet in NS-DR+ does not consider the variance of objects' physical properties when predicting dynamics.

---

> > ### Author Response · Authors · 2021-11-23
> > **Response to Reviewer DJEq (Part 2)**
> >
> > **9. Why CPL is an oracle model**
> >
> > The perception module, the program parser, and the physical property learner in CPL require explicit labels to optimize modules' parameters during training. Some pure neural network-based methods bypass physical understanding and learn correlations between questions and videos directly. Thus, they do not need such latent labels during training. We want to clarify that we need these property labels during training, which is the reason why we call CPL is an oracle model. There are some promising future directions to overcome the need for such labels during training. For example, DCL[C] and NS-DL[D] have shown that we can learn objects' dynamic and static attributes through question answering over videos and images.
> >
> > [C]. Chen Z. et al. Grounding Physical Concepts of Objects and Events Through Dynamic Visual Reasoning. International Conference on Learning Representations. 2020.
> >
> > [D]. Mao J. et al. The neuro-symbolic concept learner: Interpreting scenes, words, and sentences from natural supervision. International Conference on Learning Representations. 2019.

---

> > > ### Author Response · Authors · 2021-11-27
> > > **Eager To Receive Your Feedback**
> > >
> > > Dear Reviewer DJEq,
> > >
> > > Thanks again for your constructive comments. As the deadline for discussion is approaching, we would be glad to address any of your concerns.
> > >
> > > To address your concerns, we have made the following clarification and revisions during the rebuttal period.
> > >
> > > * We have discussed ComPhy's novelty and its difference from CLEVRER carefully in the general response 1. ComPhy V.S. CLEVRER.
> > > * We have shown ComPhy's Complexity and the idea of using more physical properties in the general response 2. About ComPhy’s Complexity and More Physical Properties.
> > > * We have also discussed the generation of training/ validation/ testing splits in the original rebuttal response and Section 3.1. We have also discussed the concern of overfitting the testing splits in the original rebuttal response. We have shown the generalization abilities of different models to a new and more complex testing split in Section 5.2 "Generalization to More Complex Scenes".
> > > * We provide more details about the CNN-LSTM model in Section 4.1.
> > > * We have also clarified your other concerns in the original rebuttal response. Such responses include the reason for the baseline model performance, generalization to unseen objects, details of Equation 2, the reason why CPL is more effective, and why CPL is an oracle model.
> > >
> > > We wish that our response has addressed your concerns and turned your assessment to the positive side. If you have any more questions, please feel free to let us know during the rebuttal.
> > >
> > > We appreciate your reviewing comments and suggestions!
> > >
> > > Best,
> > > Authors

---

> > > > ### Comment · Reviewer_DJEq · 2021-11-28
> > > > **No change in the rating**
> > > >
> > > > Thanks for the response. I read it carefully. I didn't find the response convincing due to the following reasons so I keep the initial rating:
> > > >
> > > > - I explicitly asked for a metric/statistics to show there is a significant difference between train and test scenarios. The rebuttal refers to Sec 3.1 and repeats what is mentioned there. Not being able to overfit to the test set does not mean that train and test sets are different. That can be due to the learning rates, architecture, etc.
> > > >
> > > > - The dataset is simplistic. It includes only two properties, and one of them is not really useful in downstream applications (the rebuttal does not address that comment). The rebuttal mentions "the primary goal of ComPhy is to diagnose models' physical reasoning abilities". However, it is not clear to me how a synthetic dataset with simple shapes (cubes, spheres, etc.) and only one physical property can be useful for physical reasoning in downstream real world applications.
> > > >
> > > > - Regarding the comparison with CLEVERER, the rebuttal discusses the bias of CLEVERER. My point is that they are conceptually very similar. Almost after 2 years since the publication of CLEVERER, there should be a significant difference in terms of scene realism and comprehensiveness of the physical properties.

---

### Official Review · Reviewer_3cQE · 2021-11-02

**Correctness:** 3
**Technical Novelty And Significance:** 2
**Empirical Novelty And Significance:** 1
**Recommendation:** 3
**Confidence:** 4

**Main Review:**

This paper is structured identically to CLEVRER (Yi et al., ICLR 2020): Figures 1 and 2 contains examples of scenes and questions from the dataset. Table 1 uses check marks to highlight features of existing datasets. Figure 3 shows the frequency of question types in a pie chart. Figure 4 presents sample programs representing the questions. Table 2 compares baseline methods (with almost the same column headers as the corresponding table in CLEVRER). And so on.

Beyond the writing, the models presented (CPL versus NS-DR) are also broadly the same. The similarity of the two papers raises the concerning possibility of plagiarism, or more likely, a marginal contribution sold anew in recycled packaging. Nevertheless, I will attempt to assess the work on its own merits.

**Strengths**:
- the code is public
- human baseline scores
- some attempt to avoid mistakes in past datasets (e.g. the exclusion of counterfactual questions on objects which have no interaction with other objects)

**Weaknesses**:
- Dataset: there's no innovation in the type or structure of questions asked. The generation makes arbitrary choices e.g. a video can contain a pair of charged objects (or none), or up to one heavy object. The visuals themselves are highly similar to prior work (borrowing originally from CLEVR), contributing to collective overfitting over time.
- The model: neither the CPL framework nor the implementation of any module is novel. It is entirely supervised, and exhibits little generalization.
- Writing:
  - there's several instances of overclaiming (e.g. "we compare the mass and charge edge label prediction result with the ground-truth labels and find that they achieve an accuracy of 90.4% and 90.8%. This shows the effectiveness of CPL for property identification." Given everything is trained with supervision, these prediction accuracies are no surprise.)
  - the paper is disorganized (e.g. human baselines are presented inline but not in Table 2, the generalization results in Figure 6 use a different method than in Table 3, etc). Several parts are unclear (e.g. "To run NS-DR successfully in ComPhy, we provide NS-DR with extra ground-truth physical property labels. The variant, NS-DR+ uses the PropNet (Li et al., 2019b) for dynamics predictions, which does not consider the mass and charge information of the objects." This sounds contradictory to me.)
  - it is also sparsely detailed (e.g. how exactly was the human data collected? Are the modules trained end to end or one by one?).
- Empirical results:
  - despite the numerous baselines in the paper, some important ones are missing. For instance: how would an oracle model perform if it understood all object properties but had no inkling of the latent properties (mass and charge)? That would help reveal the significance of latent properties over the remaining properties.
  - there's no error bars for any of the results.

More philosophically: do we actually need models which can infer/understand physical properties like charge from visual observation? Or models which can figure out–to use the example in the paper–why "apples float in water while bananas sink?" It doesn't seem to me that this kind of knowledge emerges from direct observation or experience in humans.

**Summary Of The Paper:**

This work extends prior datasets (chiefly, CLEVRER) for physical reasoning from visual input. It endows objects with two latent properties–mass and charge–thereby producing object dynamics like attraction or repulsion. The paper also presents VQA results on the dataset from a modular architecture (similar to NS-DR from CLEVRER) trained by supervision.

**Summary Of The Review:**

Both the presented dataset and model framework are minor tweaks of existing work. There's no novel insight. The dataset will not enable any new research directions. The paper also needs more work. I can't recommend accepting at all.

---

> ### Author Response · Authors · 2021-11-23
> **Response to Reviewer 3cQE (Part 1)**
>
> Thank you for your comments.
>
> **1. About ComPhy's Novelty and its difference with CLEVRER**
>
> Please refer to the general response **1. ComPhy V.S. CLEVRER** to show ComPhy's novelty and its difference from CLEVRER.
>
> **2. About the Writing**
>
> Thanks for your reminder on the writing. We agree that we adopt some presentation formats from CLEVRER such as using checkmarks to highlight dataset differences and comparing the performance of similar baselines. We take your reminder seriously. We will revise the presentation formats carefully and give specific references in the main text.
>
> **3. About the Novelty of Question types and Videos**
>
> **[Question Types]**. We **never** claim the question types are ComPhy's main contributions. ComPhy does have new question forms and operators for physical properties that have not been studied in CLEVRER and we have summarized the textual question templates in table 4 of the appendix. Such textual and operator differences can also be seen between Figure 7 and 8 of the ComPhy paper and Figure 2 of the CLEVRER paper.
>
> **[Videos]**. The visual primitives in ComPhy are simple and similar to CLEVRER. However, in ComPhy, we do **not** focus on the pattern recognition of diverse visual scenes but emphasize physical property inference from only a few raw video examples and property-based dynamic prediction. As shown in table 2 and table 3, all models including the CPL model with much supervision still have difficulties achieving high accuracies. This shows that ComPhy is challenging for current AI models although it uses only videos with simple visual primitives.
>
> **4. About the novelty of CPL**
>
> We agree that CPL requires explicit property labels to infer objects' properties and that is the reason why we call CPL is an oracle model.
> The core novelty of CPL lies in providing concrete solutions to 1). Physical Property Inference and 2). Physical Property-based Dynamic Prediction.
> * 1). CPL contains a graph neural network-based physical property learner (PPL) to infer objects’ physical properties from the objects’ motion and interaction in the given target and reference videos while NS-DR in CLEVRER has no such design and thus can not answer questions in ComPhy. PPL is able to infer physical properties from objects’ motions and interactions.
> * 2). The dynamic predictor of NS-DR in CLEVRER has no mechanism to model the physical properties (mass and charge) of objects in ComPhy, which leads to inferior dynamic prediction performance. Instead, CPL has specific modeling for mass and charges on nodes and edges of the graph neural network and achieves better performance.
>
> **5. About the Overcliaming of the Effectiveness of CPL for Property Identification**
>
> Our oracle model CPL requires the property labels to train the graph neural networks to infer the property labels. By claiming that "This shows the effectiveness of CPL for property identification", we would like to show that  graph neural networks are able to learn physical properties from objects' trajectories and interactions in a supervised manner. We will provide more accurate description of the performance in the revision.
>
> **6. About the Organization of the Paper**
>
> **[About baseline comparison]**. We compare CPL and NS-DR+ in table 3 since we want to show the importance of predicting physical-based dynamics. We choose ALOE and ALOE(Ref) as the baselines to compare with CPL on more complex scenes since ALOE and ALOE(Ref) are the best baselines shown in table 2.
>
> **[About details of NS-DR+]**. NS-DR requires property labels to execute the programs of language but makes dynamic predictions without considering physical property value variance. For example, given the question “If the purple sphere were heavier, what would happen?”, NS-DR needs to know the mass value of the “purple cube” to execute the program operator ("Counterfact heavy"). For better analysis, we implement a variant of NS-DR with extra ground-truth physical property information to execute the programs. By default, NS-DR uses PropNet for dynamic predictions. The PropNet does not consider the variance of property values. The reviewer may refer to the CLEVRER paper for more details on program execution and dynamic prediction.
>
> **[Other Details]**. We will provide more details of human evaluation in the Paragraph of **Human Performance** of Section 4.2 and update them in table 2 in the revised version. Modules in ComPhy are trained one by one and more details are provided in Section D of the Appendix.

---

> > ### Author Response · Authors · 2021-11-23
> > **Response to Reviewer 3cQE (Part 2)**
> >
> > **7. About Oracle Models without object properties**
> >
> > We have shown a CPL model without knowing objects' physical property labels (mass and charge) and it achieves only an accuracy of 55.2% on factual questions, which is much lower than 80.5% in table 3. Also, the PropNet-based dynamic predictor in NS-DR+, which has no consideration of property labels for dynamic predictions, has worse performance than the CPL model, which encodes physical properties during dynamic prediction. These results show the importance of physical properties in ComPhy.
> >
> > |          | CPL      |  CPL w/o physics |
> > | -------- | -------- | -------- |
> > |Factual   | 80.5     | 55.2     |
> >
> > **8. About the Error bars of Experimental Results**
> >
> > Thanks for the reviewer's suggestion on proving error bars for experimental results. We do not calculate the error bars multiple times for the following reasons. First, ComPhy is large enough to provide stable and deterministic results. Second, there is much effort in running all the baselines multiple runs to calculate the error bars since it may take days to run a single baseline like ALOE once. Related previous benchmarks like CLEVR, CLEVRER, and Cater did not provide error bars for all the baselines either. However, they still provide stable and reliable results and make strong impact in the research area.
> >
> > **9. About the Significance of Understanding Physical  Properties from Visual Observation**
> >
> > **[Significance to understand Physics]**. We believe it is significant to understand physical properties from visual observation. Objects in nature often exhibit complex physical properties and these properties have a strong impact on objects' motions and interaction. The understanding of these physical properties and their effect is of great importance since they may provide humans with a deeper and more accurate understanding of the physical world around us. And it can also be applied to real-world robotic control and planning as described in [A] and [B].
> >
> > **[About the Example]**. The reviewer claims that knowledge that "why apples float in water while bananas sink" does not emerge from direct observation of humans. However, by observing "apples float in water while bananas sink", humans can infer apples' density (mass per unit volume) is smaller than that of bananas. This is an example to support our claim in ComPhy that we can infer objects' physical properties (density) from objects' movement and interactions ("float" and "sink").
> >
> > [A]. Veerapaneni, Rishi, et al. "Entity Abstraction in Visual Model-Based Reinforcement Learning." CoRL (2019).
> >
> > [B]. Janner, Michael, et al. "Reasoning about physical interactions with object-oriented prediction and planning." ICLR (2019).

---

### Official Review · Reviewer_8BUA · 2021-11-04

**Correctness:** 3
**Technical Novelty And Significance:** 3
**Empirical Novelty And Significance:** 3
**Recommendation:** 6
**Confidence:** 4

**Main Review:**

### Strengths:

1. The task definition, formulation, and the proposed dataset Comphy are novel and thoughtful. Models working on other physics-based (such as Phyre, Cleverer, Cophy) datasets may or may not work on this dataset (Comphy).

2. Multiple baselines are tested and the proposed model CPL shows significant gain over these.

### Weakness:

1. On Motivation: The paper doesn't motivate the problem, and I am struggling to see any direct application. Could the authors provide any examples where such a model would be useful?

2. (Minor): In Table 1., the authors suggest CoPhy and Cleverer do not support counterfactual, but to my understanding they do? I could be missing something here.

3. On Task and dataset setup:

(i) For intrinsic properties, the authors chose mass and charge suggesting that inferring the two is challenging. However, this is not explicitly shown in experiments. Authors should show that just finding mass or just finding charge from reference videos is doable, but having both in properties together makes the task difficult (I believe this to be the case intuitively, but an explicit expt would be helpful).

(ii) The reason for choosing 4-videos as reference is unclear as compared to directly providing the values as input. For instance, the input could itself be the target video + properties of the objects, followed by multiple questions.

(iii) How is number "4" for reference videos chosen? Does the task become simpler for a particular target video if more reference videos are added (say 10 or 1000)?

(iv) In the current setup, each target video has 4-reference videos, followed by a number of questions. A simple extension would be to use the same 4-reference video, but use a different target (same objects, same properties but different target videos). This could suggest if the model is is able to answer some questions by chance or did it actually infer the physical properties.

4. On Model and Experiments:

(i) Intermediate results: Baselines showing results with ground-truth object properties should be reported. For instance, NS-DR and CPL in Table2., but  they are given the ground-truth heavy/light information as well as the corresponding charge. (This is related to point 3.(ii) ).

(ii) The task of deriving object properties from the 4-reference video is independent of the questions asked on target video. As such, the authors should report separate metrics for just this part. This would show if future research should focus on deriving physical properties or given physical properties work on the counterfactual/predictive types of question.

(iii) The physical property inference is (to the best of my understanding) is applied on 4 reference videos + target video. This setup of being dependent on target video seems confusing to me. As such a model which has seen 4 reference videos should be able to work on a completely new target video as well (see related point 3.(iv)) Could the authors show results without using target video?

**Summary Of The Paper:**

The authors propose a new task and corresponding dataset ComPhy, a video question-answering dataset for evaluating video reasoning capability. First, a model is given 4 reference videos (2-sec clips showing object interactions in isolation) which is used to deduce object properties. Then a new query video (which could have more than 2 objects) is given on which 3 types of questions are asked: factual (about what happens in the video), counterfactual (what would happen if some condition changed), and predictive (what would happen after), with later two being multiple-choice questions. A particular contribution is the focus is on latent properties of the objects (mass and charge) which needs to be deduced from the reference videos. Multiple video reasoning model baselines are reported along with a newly proposed oracle Composition Physics Learner (CPL) model which shows considerable improvement over the baselines.



**Summary Of The Review:**

The authors provide a novel and well-thought task and corresponding dataset ComPhy which requires a model to use 4-reference videos to infer intrinsic properties of the objects and thereby apply these findings to a target video and answer corresponding factual/predictive and counterfactual questions. However, in its current form, the exact motivation of the paper is unclear, the reliance on using 4-reference videos compared to simply providing the information is not justified, and some dataset choice such as using only target video for given reference videos are not clear. More intermediate results such as finding object properties would make the paper stronger.

---
Post Rebuttal: The authors have substantially updated their paper with additional ablative studies and human evaluation results. In my opinion, this substantially improves the findings and takeaways of the paper. More experiments with physical properties in isolation, showing the effect of increasing the number of reference videos, and de-coupling the reference and target videos could further strengthen the paper.

As such, I am increasing my score to 6.

---

> ### Author Response · Authors · 2021-11-23
> **Response to Reviewer 8BUA (Part 1)**
>
> Thank you for your constructive comments.
>
> **1. Applications of the CPL model**
>
> CPL is a neural-symbolic framework, which is a modularized model that can infer objects’ physical properties and predict the objects’ movements. CPL's Core modules are object-centric graph neural networks that capture the compositional physical properties of the underlying system. Such object-centric framework for physical reasoning and dynamic predictions have been proved effective in [A] and [B] for robotic planning and manipulation in the real world.
>
> [A]. Veerapaneni, Rishi, et al. "Entity Abstraction in Visual Model-Based Reinforcement Learning." CoRL (2019).
> [B]. Janner, Michael, et al. "Reasoning about physical interactions with object-oriented prediction and planning." ICLR (2019).
>
> **2. About Support of Counterfactual Situations**
>
> Thanks for pointing out that Cophy and CLEVRER support counterfactual reasoning in table 1. In table 1, what we are trying to point out is "**Counterfactual Property-based Dynamics**" rather than "**Counterfactual dynamic**". Counterfactual reasoning in CoPhy and CLEVRER has nothing to do with the specific physical property (mass and charge). CoPhy requires models to predict counterfactual dynamics under different object initialization positions. CLEVRER requires models to predict dynamics under the hypothesis that an object is removed. Differently, ComPhy asks counterfactual questions under the hypothesis that a certain object carrying a different physical property like "If the sphere were **uncharged**, what would happen?".
>
> **3. About Physical Property Inference**
>
> We claim that it is challenging to infer physical property labels (mass and charge) since existing models fail to achieve reasonable results on ComPhy. We have shown the results of CPL on physical property identification in "Effectiveness of Physical Property Learning" of Section 5.2. We compare the CPL's mass and charge edge label prediction result with the ground-truth labels and find that the GNNs in CPL achieve an accuracy of 90.4% and 90.8%. This shows that in ComPhy it is doable to infer objects' physical properties from the target video and the references with explicit graph neural networks and property labels during training. The performance gap between property label inference and overall question-answering performance shows that the bottleneck of the CPL model lies in physical based dynamic prediction.
>
> **4. Reasons for choosing 4 reference videos rather than directly providing the property labels or more reference videos**
>
> **[Why use reference videos rather than ground-truth property labels]**. If we directly give the property labels of objects, the task will be simpler and we can still evaluate models' ability to predict dynamics based on objects' physical properties. However, besides the goal to make physical-based dynamics prediction, another important goal of ComPhy is to evaluate models' ability to learn the hidden physical properties from objects' motion and interactions of only a few examples. Reference videos can provide important information for models to infer physical properties. It also makes our task closer to real-world applications in that we infer objects' properties from motion and interactions in raw videos rather than directly knowing the property labels.
>
> **[Why use only 4 reference videos]**. We choose 4 videos since we want to provide enough interaction information (collision, attraction, and repulsion) to the models. As shown in the table of the following point **6. Effectiveness of reference videos**, we find that 1). we can infer objects' physical properties from the given videos; 2). 4 reference videos provide important information to infer the objects' properties. It will be easier for the models to infer physical properties when more reference videos (10 or 100) are added since they provide more interactions among objects. However, it will be much more computation-intensive for models to analyze these reference videos one by one. We want the models to be efficient since humans can also efficiently infer objects' physical properties from only a few video observations.
>
> **5. About adding additional target videos**
>
> Thank you for your suggestion that "using the same reference videos with a different target to evaluate whether models answer questions by chance or it actually infers the physical property". We have explicitly shown the accuracy of physical property in "Effectiveness of Physical Property Learning" of Section 5.2 and the following **6. Effectiveness of reference videos**. We may consider your suggestion in the later version.

---

> > ### Author Response · Authors · 2021-11-23
> > **Response to Reviewer 8BUA (Part 2)**
> >
> > **6. Effectiveness of reference videos**
> >
> > **[Reasons to Use the Target Video]**. We use both the target video and reference videos rather than only the reference videos for physical property inference since **we need to provide a target video to the model for video question answering**. Note that pure neural network baselines in Table 2 do not explicitly infer the physical properties and directly predict the answers based on the given target video, reference videos, and question description. We believe it is a more reasonable setting for models to use all the provided video inputs rather than only the reference videos to infer the physical properties in the proposed CPL model.
> >
> > **[Effectiveness of Reference Videos]**. To evaluate the importance of reference videos for physical property inference, we implement a baseline graph neural network that only relies on the target video without reference videos. It achieves an accuracy of 59.4% and 70.2% on mass and charge edge labels, which is much smaller than the accuracy of 90.4% and 90.8% described in "Effectiveness of Physical Property Learning" of Section 5.2. We will update such experimental analysis in the revision.
> >
> > |                | Mass     | Charge   |
> > | --------       | -------- | -------- |
> > |w/o reference   | 59.4%     | 70.2%   |
> > |   Full         | 90.4%     | 90.8%   |
> >
> > **7. Baselines with ground-truth property labels**
> >
> > For better analysis, we have also reported a baseline with ground-truth object properties and visual static and dynamic attributes. We show the performance of the ground-truth baseline in the following table. Since we use the ground-truth annotation, it achieves perfect performance on factual questions. For questions in counterfactual and predictive scenes, although it achieves constant gains, there is still much room for further improvement. This shows the bottleneck of ComPhy lies in predicting physical-based dynamics on ComPhy.
> >
> > |                 | Factual  | Pred. Per Opt.   |Pred. Per Opt.   |Count. Per Opt.   |Count. Per Opt.   |
> > | --------     | -------- | -------- |-------- |-------- |-------- |
> > |CPL          | 80.5%    | 75.3%    | 56.4%   | 68.3%   |  29.1%  |
> > |GT label    | 100%     | 87.6%    | 74.0%   | 74.0%   |  35.3%  |

---

> > ### Comment · Reviewer_8BUA · 2021-11-27
> > **Thanks for the detailed response**
> >
> > 1. On Application: My comment was more on the application of the task. As in suppose, there is a black-box method that can solve the ComPhy task, where can it be used?
> >
> > 2. About Support of Counterfactual Situations: I get it now, but I feel the writing can be improved to drive home this point.
> >
> > 3. About Physical Property Inference:
> >
> > I am unclear of the motivation behind the design choice of using both mass and charge in physical property To clarify, one needs to show that finding out just "mass" and just "charge" i.e. keeping one physical property fixed and the other varying is "doable" while trying to find out both is "not doable". Otherwise, it makes more sense to have just one physical property (mass or charge).
> >
> > 4. Reasons for choosing 4 reference videos rather than directly providing the property labels or more reference videos
> >
> > To me, it seems the two tasks are independent, and ideally one should compute three things: (a) ability to predict physical properties from videos, (b) ability to reason given physical properties and target video, and (c) the currently proposed task of ComPhy. Essentially, (a), (b) break down (c) and give us more insights about where the bottleneck lies.
> >
> > I believe the edge labels (as provided in point 6) are indicative of the performance of (a), and results in point 7. are indicative of (b). However, more results on (a) should be shown such as using 4 videos vs 10 videos vs 100 videos (perhaps even for a subset of examples).
> >
> > ---
> > On the whole, the new results especially point 6, 7 as well as the human performance (Table 2) improve the paper and its findings. As such I am raising my score to 6.

---

> > > ### Author Response · Authors · 2021-11-27
> > > **Thanks again for the valuable feedback.**
> > >
> > > Dear reviewer 8BUA,
> > >
> > > Thanks for your valuable updated comments. And we further address your remaining concerns as follows.
> > >
> > > **1. The Goal and Applications of the Task**
> > >
> > > **[Applications of Physical Reasoning]**. We first emphasize that physical reasoning is important for agents to perceive the environment and perform downstream tasks. For example, under the same push, the lighter object will likely move a longer distance than a heavier object, but such property may not be directly observable from still images. Therefore, being able to infer the invisible object properties from videos is of critical importance to build a more accurate model of the environment (e.g. [A]) and facilitate downstream tasks like object manipulation (e.g. [B]). We agree that physical reasoning in the real world is complex and difficult. That is also the reason why we build ComPhy, a simple but representative benchmark, to evaluate AI models' abilities for physical reasoning.
> > >
> > > **[ComPhy's Goals]**. The goal of the tasks in ComPhy is to diagnose models' abilities for physical reasoning. The synthetic nature of ComPhy allows us to control the distribution of scenes, questions, and answers. Moreover, we can diagnose models with ground-truth physical states and programs. More generally, our dataset represents a perspective that synthetic and diagnostic datasets will enable progress in machine reasoning. It is a perspective that many in the community hold (e.g. CLEVR[Johnson et al., 2017] and CLEVRER[Yi et al., 2020]). Such diagnostic benchmarks like CLEVR and CLEVRER have made a strong impact in the field.
> > >
> > > **[About Transforming to Real-world Applications]**. We agree that much effort is needed for such result systems to transfer to real-world applications (e.g. a better perception module to detect object attributes and a stronger language parser to parse natural language).
> > >
> > > [A]. Fabio, Ramos, et al. BayesSim: Adaptive Domain Randomization Via Probabilistic Inference for robotics simulators. RSS, 2019.
> > >
> > > [B]. Chebotar, Yevgen, et al. Closing the Sim-to-Real Loop: Adapting Simulation Randomization with Real World Experience. ICRA. 2019.
> > >
> > > **2. About Support of Counterfactual Situations**
> > >
> > > Thanks for your advice on the writing of the counterfactual situations and we will update the revised paper carefully.
> > >
> > > **3. About the Property Inference**
> > >
> > > **[Clarification]**. Thanks again for your concerns on using both mass and charge in physical properties. We respectfully think that the reviewer might misunderstand the property setting in ComPhy. Compositional Physical Reasoning in ComPhy means that the composition between mass (or charge) with the visible attributes (color, shape, and material) rather than the co-occurrence between the mass and charge. **We will further clarify and update these details in the revised version.**
> > >
> > > **[Charges and Mass in ComPhy]**. As discussed in Section 3.3 BALANCING AND STATISTICS, we manually make sure that a video set will only contain a pair of charged objects if it contains charged objects. Similarly, a video will only contain a heavy object or no heavy objects. To make it simple for humans, we simplify the situations in ComPhy and there are **no videos** (expect videos in the generalization split) currently in ComPhy that contain both charged objects and heavy objects.
> > >
> > > **[Generalization to More Complex Situations]**. As discussed in Section 5.2 "Generalization to More Complex Scenes", we have an additional testing split to test the model's generalization abilities. Videos in the additional split have a larger object number(6-8) and contain both charge objects and heavy objects. Such more complex videos are more challenging. The proposed CPL model and the previous STOA baseline ALOE both achieve worse performance on this generalization split. We will provide a more detailed analysis in the later version according to the reviewer's comment.
> > >
> > > **4. About More Reference Videos**
> > >
> > > Thanks for your suggestions on more ablation studies on reference videos (e.g. 4 videos vs 10 videos vs 100 videos). We will provide a more detailed ablation study in the later version to study the effect of the number of reference videos.
> > >
> > > Thanks again for the valuable feedback!
> > >
> > > Best,
> > > Authors

---

### Official Review · Reviewer_VByS · 2021-11-04

**Correctness:** 3
**Technical Novelty And Significance:** 3
**Empirical Novelty And Significance:** 3
**Recommendation:** 6
**Confidence:** 4

**Main Review:**

This paper is overall well written and executed. With a focus on inferring hidden attributes, this work tackles a crucial aspect of visual reasoning. Experiments are well-designed to illustrate the claimed contributions.

My main concerns are about the complexity that the proposed method can handle. Overall, the method is only demonstrated on a simple synthetic dataset with two attributes (mass and charge), while mass only has two values (heavy vs light). Despite it does demonstrate the claimed contributions, to what extent the proposed method would work and generalize is unknown.

Additionally, it seems that all objects in a video in the datasets are of interest; there are no irrelevant objects. Would this become a problem for the algorithm?

For a visual reasoning task, the proposed method seems to struggle in the following scenario: If A and B are attracted, and B and C are attracted, A and C should be repelled.

**Summary Of The Paper:**

This paper proposes a dataset and a method to study the hidden properties of objects present in videos. The method can extract object attributes and use them for predicting subsequent scenes, with the additional capability to handle counterfactual scenarios.

**Summary Of The Review:**

This is a good work that fills the gap of visual reasoning of hidden attributes, but experiments could be more comprehensive.

---

> ### Author Response · Authors · 2021-11-23
> **Response to Reviewer VByS**
>
> Thanks for your detailed review.
>
> **1. About the Capacity and Generalization of CPL model**
>
> **[Capacity]**. The proposed CPL model takes the first step to infer objects' hidden physical properties from raw video observations and predict objects' dynamics based on the estimated physical properties. Currently, CPL shows some promising results on this very challenging problem. Though the visual primitives in ComPhy are simple, we would like to emphasize that models (including the proposed CPL) are still far from solving the problem.
>
> **[Generalization]**. At the core of our CPL model is object-centric graph neural networks for physical property learning and dynamic prediction. As shown in [A] and [B], such object-centric physical reasoning and dynamic predictions have been generalized to real-world applications like robotic planning and manipulation.
>
> [A]. Veerapaneni, Rishi, et al. “Entity Abstraction in Visual Model-Based Reinforcement Learning.” CoRL (2019).
>
> [B]. Janner, Michael, et al. “Reasoning about physical interactions with object-oriented prediction and planning.” ICLR (2019).
>
> **2. About the Effect of Irrelevant Objects for CPL Model**
>
> **[Irrelevant Objects]**. As described in Section 3.1 Videos, there are no "irrelevant" objects that have **not** appeared in the target video since objects in reference videos are sampled from the target video with different initial velocities and locations. Each object can be uniquely identified by the combination of the three attributes (color, shape, and material).
>
> **[Irrelevant Objects' Effect]**. As described in the "Physical Property Inference" paragraph of Section 5.1, CPL uses graph neural networks to independently predict node property (mass) and edge property (charge) for each reference or target video. CPL then aligns objects' property predictions in different videos by the predicted static attributes from the perception module. CPL gets the final predictions of each object by max-pooling over the prediction of aligned nodes or edges. If there is an "irrelevant" object (an object with different static attributes) in reference videos, there will be an "irrelevant" node in that video graph and our CPL still works well.
>
> **3. About the CPL MODEL for Situation of Multiple Charged Objects**
>
> Thanks for pointing out the interesting reasoning situation that ": If A and B are attracted, and B and C are attracted, A and C should be repelled.". CPL adopts a multi-layer graph neural network (GNN) to predict objects' physical properties based on objects' trajectories. Ideally, we expect the GNN to learn objects' relation (the repulsion of A and C in the mentioned example) through multi-step message propagations.

---

> > ### Author Response · Authors · 2021-11-27
> > **Look forward to your feedback!**
> >
> > Dear reviewer VByS,
> >
> > Thanks again for your feedback. As the deadline for discussion is approaching, we would be glad to provide any additional clarifications that you still need.
> >
> > According to your initial review, we have made the following clarification in the previous comments.
> >
> > * We discuss the capacity and generalization of the proposed CPL model in the previous comment;
> > * We have clarified the "irrelevant objects" in ComPhy and discussed its effect on the model;
> > * We have discussed the abilities of the GNN in CPL to handle the interesting reasoning situations that you mentioned.
> >
> > If you still have any remaining concerns, we would be happy to do anything that we can that would be helpful in the time remaining!
> >
> > Thank you again for your effort and time in reviewing this paper.
> >
> > Best,
> > Authors

---

### Author Response · Authors · 2021-11-23
**General Response to all reviewers (Part 1)**

We sincerely thank all reviewers and ACs' time and efforts in reviewing the paper. We are glad that reviewers recognized the following contributions.
* **Task**. The task is novel and well-motivated ("a crucial aspect of visual reasoning"(VByS), "Novel and thoughtful" (8BUA), "a quite interesting direction" (DJEq)).
* **Experiments**. Experiments are well-designed ("experiments are well-designed to illustrate the claimed contributions" (VByS), "multiple baselines are tested" (8BUA), ).
* **Model**. The CPL model is effective ("shows significant gain " (8BUA) "outperforms a number of baseline approaches." (DJEq)).

**1. ComPhy V.S. CLEVRER**

**[Clarification]**. We find that reviewers misunderstood our paper's novelty and would like to reemphasize this paper's main contributions. CLEVRER is an impressive and pioneering dataset that opened up an entirely new field of physical reasoning. However, there are many limitations in CLEVRER such as dataset bias, no physical property variance, and no need to understand physics to achieve high performance. We saw the need for creating a new dataset that to embrace all existing features but overcome the limitations in CLEVRER.

**[Challenges of Task Setting for Physical Reasoning]**. The key challenge is how to design a task for compositional physical reasoning and avoid shortcuts between physical properties and static attributes. A straightforward way is to follow the setting in CLEVRER, requiring the model to watch a video and then answer questions about physical properties. However, physical properties are complicated and often can not be fully unraveled in only one video. Another solution is to correlate object appearance with physical properties like making all red spheres to be heavy and then ask questions about their dynamics. However, such a correlated setting may incur shortcuts for models by just memorizing the appearance prior rather than understanding coupled physical properties.

**[Novelty of the Few-Shot Learning Setting]**. Instead, ComPhy introduces a few-shot reasoning setting for physical property learning. ComPhy first provides only a few video examples for models to identify objects’ physical properties and then asks questions about the physical properties and dynamics in the target video. Such a setting is natural since people also often infer objects' physical properties from only a few observations. Such a few-shot setting decorrelates physical properties from visual appearance and avoid shortcuts between them. We believe such a new setting is a correct way to evaluate physical reasoning. We also propose a neural symbolic framework that uses graph neural networks to infer objects' properties and predict objects' dynamics.

**[Similarity as Synthetic Datasets]**. As described in Section 3, similar to CLEVRER, we adopt a physical engine (PyBullet) to simulate physics, a graphics engine (Blender) to render dynamic videos, and a question engine to synthesize question-answer pairs with functional programs. We **never** claim that such a pipeline to generate the ComPhy dataset is our contribution. In contrast, we use the same synthetic pipeline to overcome CLEVRER' limitations and perform real physical reasoning.

**[CLEVRER's Limitations]**. Despite CLEVRER's significant contribution to physical reasoning, there are still some limitations.
* **Bias**. Some questions have specific bias and requires no need for counterfactual reasoning. For example, as pointed out by Ding et al. (2020), about half of the counterfactualobjects in CLEVRER have no collisions with other objects and the counterfactual questions can besolved merely based on the observed target videos.
* **No Physical Property Variance**. Second, objects in CLEVRER are designed with the same physical properties (insulators with the same mass value). It mainly focuses on visible properties and dynamics **without physical property variance**.
* **High Performance without Physics Understanding**. The pure neural network model, ALOE, achieves excellent accuracy on CLEVRER **without understanding physical properties**, which raises a doubt on CLEVRER's ability to evaluate physical reasoning.

Such limitations and experimental results on CLEVRER indicate strong demand on a new dataset that can incooperate CLEVRER's existing features and better evaluate models' physical reasoning abilities.

---

> ### Author Response · Authors · 2021-11-23
> **General Response to all reviewers (Part 2)**
>
> **[ComPhy's New Features]**. ComPhy aims to fill these gaps by bringing the following new features.
> * ComPhy introduces physical property variance (heavy vs light, charged vs uncharged) and reference videos into the dynamic scenes. It requires models to identify **intrinsic physical properties** of objects from **only a few video examples**;
> * ComPhy requires models to make **physical property-based dynamic predictions** for the target video like *"Which event would happen if the purple object were **heavier**?"*. As shown in Figure 1 of the main paper, the variance in physical properties (mass and charge) plays an important role in objects' dynamic predictions;
> * We provide further control to reduce the dataset biases like removing counterfactual questions on objects that have no interaction with other objects.
>
> **[New result Finding**]. As a result, the previous model ALOE, which achieved excellent accuracy (0.875 on predictive questions) on CLEVRER **without understanding physical properties**, performs poorly on ComPhy (0.371 on predictive questions). This shows the difference between CLEVRER and ComPhy and the importance of modeling physical properties on ComPhy. We made much effort to build the ComPhy dataset and we believe that such a new task setting is a correct direction for physical reasoning.
>
> **2. About ComPhy’s Complexity and More Physical Properties**
>
> **[Supported Features]**. ComPhy supports reasoning over the compositional visible and hidden physical properties. Theoretically, we can add more physical properties like bounciness coefficients and friction into the benchmark and make the values of these physical properties continuous. We can also simulate more complex scenes with advanced simulators such as AI2-THOR and iGibson.
> **[Design Logic]**. However, the primary goal of ComPhy is to diagnose models' physical reasoning abilities rather than simply building a complex dataset. We want to keep the dataset simple for people to infer the physical properties from only a few observations while still challenging for current AI models. As shown in Table 3, the oracle model CPL still achieves limited performance (56.4 on predictive questions and 29.1 on counterfactual questions) on ComPhy even though it uses much supervision information during training. We can add more other physical properties and more complex object appearance and interaction into ComPhy if the AI models can achieve satisfactory performance on the current benchmark.
>
> **3. New Experiments**
>
> In addition to the pointwise responses below, we summarize supporting experiments added in the rebuttal according to reviewers’ suggestions.
> * Ablation study to show the effectiveness of reference videos (8BUA);
> * Oracle model that uses ground-truth visual attributes and physical property labels (8BUA);
> * Oracle CPL model without knowing objects’ physical property labels (mass and charge) (3cQE).

---

### Author Response · Authors · 2021-11-26
**Thanks for all your comments and look forward to post-rebuttal feedbacks!**

Dear AC and all reviewers:

Thanks again for all of your constructive suggestions, which have helped us improve the quality and clarity of the paper!

Since the discussion phase has started for over one week, we have not heard any post-rebuttal response yet.

Please don’t hesitate to let us know if there are any additional clarifications or experiments that we can offer, as we would love to convince you of the merits of the paper. We appreciate your suggestions. Thanks!

Best,
Authors

---

### Decision · Program_Chairs · 2022-01-20

**Decision:**

Accept (Poster)

**Comment:**

This paper proposes a new dataset called ComPhy to evaluate the ability of models to infer physical properties of objects and to reason about their interactions given these physical properties. The paper also presents an oracle model (named oracle because it requires gold property labels at training time) that is modular and carefully hand designed, but shows considerable improvement over a series of baselines. The reviewers for this submission had several concerns including:
(a) [VByS] "concerns are about the complexity that the proposed method can handle"\
(b) [VByS] "the method is only demonstrated on a simple synthetic dataset"\
(c) [8BUA] "I am struggling to see any direct application"\
(d) [8BUA] "choosing 4-videos as reference" -- why use ref videos, why use 4\
(e) [8BUA] "Baselines showing results with ground-truth object properties should be reported"\
(f) [3cQE] "no innovation in the type or structure of questions asked"\
(g) [3cQE] "neither the CPL framework nor the implementation of any module is novel"\
(h) [DJEq] "The only difference is that this paper infers hidden properties instead of collisions"\
(i) [DJEq] "The dataset is not comprehensive enough" -- only 2 properties and simplistic and synthetic videos\

The authors have provided detailed responses to these concerns and I discuss these below.

The authors have addressed (c),(d) and (e) well in their rebuttal.

I don't think (a) is concerning. The proposed model is not expected to solve the dataset entirely inspite of having access to gold properties at training time. As the authors mention, this indicates the complexity of the task at hand.

The authors also address (f) well. I dont think there is any need for innovation in the structure of questions asked. QA is merely a mechanism to probe the model, and using CLVERER style questions seems appropriate.

I disagree with the sentiment behind (g). The proposed oracle model clearly inherits modules from past works and assembles them to suit the needs of the dataset. It is this assembly that differentiates it from past works. This is true of most papers in our field, including ones that are widely acknowledged to be important papers. The underlying modules in proposed networks are rarely novel, but their assembly can lead to improvements on benchmarks. Furthermore, the oracle model, isnt the central contribution of this work. The dataset is, and hence, the requirement for novelty is reduced. The oracle is meant to serve as a guideline to show what one may achieve given gold labels at training, and it serves that purpose well.

Re (h), my takeaway is that inferring properties based on their dynamics and without any link to their appearance is an important step, and past datasets do not exhibit this characteristic. And thus, in spite of being a limited differentiation from CLEVERER, I think this is interesting.

Re: (b) and (i) I do agree with some aspects of these, with the reviewers.
I think its still valuable to have a dataset with synthetic videos, given that models today are unable to solve this dataset. Moving to more realistic videos is a next step.
However, as the reviewer [DJEq] points out, it would be desirable to add more physical properties and add more complex scene elements like ramps. That would have added a lot more diversity to the dataset -- visually, with regards to physical properties and with regards to the types of reasoning required.

Having said that, I believe that the dataset in its present form is still valuable to the community, and hence I recommend acceptance.
I think adding more physical properties and scene elements will have made this a much stronger submission.